

# LUCIE-3D: A three-dimensional climate emulator for forced responses

Haiwen Guan[1], Troy Arcomano[2,3], Ashesh Chattopadhyay[4], and Romit Maulik[1,3]

[1]College of Information Sciences and Technology, Pennsylvania State University, University Park, PA, USA
[2]Allen Insitute of Artificial Intelligence, Seattle, WA, USA
[3]Argonne National Laboratory, Lemont, IL, USA
[4]Applied Mathematics, University of California, Santa Cruz, CA, USA

**Correspondence:** Haiwen Guan (hzg18@psu.edu)

**Abstract.** We introduce **LUCIE-3D**, a lightweight three-dimensional climate emulator designed to capture the vertical structure of the atmosphere, respond to climate change forcings, and maintain computational efficiency with long-term stability. Building on the original LUCIE-2D framework, LUCIE-3D employs a Spherical Fourier Neural Operator (SFNO) backbone and is trained on 30 years of ERA5 reanalysis data spanning eight vertical $\sigma$-levels. The model incorporates atmospheric $CO_2$

as a forcing variable and optionally integrates prescribed sea surface temperature (SST) to simulate coupled ocean–atmosphere dynamics. Results demonstrate that LUCIE-3D successfully reproduces climatological means, variability, and long-term climate change signals, including surface warming and stratospheric cooling under increasing $CO_2$ concentrations. The model further captures key dynamical processes such as equatorial Kelvin waves, the Madden–Julian Oscillation, and annular modes, while showing credible behavior in the statistics of extreme events. Despite requiring longer training than its 2D predecessor,

LUCIE-3D remains efficient, training in under five hours on four GPUs. Its combination of stability, physical consistency, and accessibility makes it a valuable tool for rapid experimentation, ablation studies, and the exploration of coupled climate dynamics, with potential applications extending to paleoclimate research and future Earth system emulation.

## 1 Introduction

Machine learning (ML) has brought substantial advancements to atmospheric science in recent years. In particular, several

ML-based models for medium-range weather prediction have demonstrated forecasting skill comparable to, and in some cases surpassing, traditional numerical weather prediction (NWP) systems (Pathak et al., 2022; Lam et al., 2023; Bi et al., 2023; Chen et al., 2023). Building on this momentum, climate modeling has also seen rapid progress through the integration of ML approaches. Unlike short-term weather forecasting, which often becomes unstable or unphysical beyond the two-week predictability horizon, climate emulators must maintain long-term stability to capture reliable statistics and climate dynamics over

decades (Chattopadhyay and Hassanzadeh, 2023), while also accurately responding to changes in radiative forcing that drive long-term climate change. This requirement introduces unique challenges for data-driven models. Although a number of recent efforts have shown promise in addressing these constraints, e.g., Neural GCM (Kochkov et al., 2024), DlESyM (Cresswell-Clay et al., 2025), and LUCIE-2D (Guan et al., 2024) (developed by us), only ACE2 (Watt-Meyer et al., 2025) and CAMula-





tor (Chapman et al., 2025) have thus far demonstrated the ability to robustly capture long-term climate change signals. It must
be noted, though, that the major difference between CAMulator, ACE2, and LUCIE-2D/LUCIE-3D (this work) is that the latter
two are trained on observation-derived reanalysis data while CAMulator is trained on climate model simulations.

Another critical challenge in building such emulators is the overall computational cost associated with their development
and testing. While ML-based climate emulators are far more efficient than conventional NWP systems during inference, their
training phase can demand extensive computational resources, creating barriers for many researchers. To lower this entry
barrier, the original LUCIE-2D framework (Guan et al., 2024) was developed as a lightweight climate emulator designed to
minimize both computational overhead and training data requirements. Similar efforts can be seen in Cresswell-Clay et al.
(2025) as well. By prioritizing efficiency, LUCIE-2D enabled broader accessibility for exploring and prototyping novel model
architectures. However, as a minimal working model, LUCIE-2D was limited to a subset of vertical levels, restricting its ability
to fully represent the three-dimensional dynamics of the climate system.

In this paper, we present **LUCIE-3D**, a data-driven climate emulator that extends the LUCIE-2D framework to explicitly
incorporate the full vertical structure of the atmosphere. The model continues to leverage the Spherical Fourier Neural Operator
(SFNO) as its backbone while maintaining the efficiency-oriented design of LUCIE-2D. Unlike its predecessor, which was
trained on a limited number of $\sigma$-levels, LUCIE-3D is trained on data spanning the full vertical extent of the atmosphere.
This vertical coverage proves essential for capturing key dynamical processes such as equatorial Kelvin waves. Furthermore,
LUCIE-3D introduces atmospheric $CO_2$ as a forcing variable, enabling the emulator to reproduce climate change signals, and
includes a variant trained with sea surface temperature (SST) as a forcing to assess its potential for AMIP-style simulations.
Despite handling an expanded set of 34 variables over 30 years of ERA5 data, LUCIE-3D remains computationally efficient,
requiring fewer than five hours of training on four A-100 GPUs.

The long-term stability, physical consistency, and lightweight nature of LUCIE-3D make it a practical tool for rapid experi-
mentation with architectures, loss functions, and training strategies. Its efficiency further enables systematic ablation studies to
evaluate the influence of different atmospheric and forcing variables on climate dynamics. Importantly, the model shows strong
potential for integration with oceanic components, paving the way for the development of a fully coupled climate emulator.

The remainder of this paper is organized as follows: section 2 introduces the ERA5 dataset used for training; section 3
describes the model architecture, loss function, and training strategy; section 4 evaluates the model's ability to reproduce
observed climatology and variability, responses to external forcings ($CO_2$ and sea-surface temperature perturbations), and spin-
up from arbitrary out-of-distribution initial conditions; and section 5 discusses limitations, implications for future development,
and potential applications.

## 2 Dataset

In this study, we employ the ERA5 reanalysis dataset, regridded to a T30 Gaussian grid and vertically interpolated across eight
$\sigma$-levels (terrain following coordinates) (Arcomano et al., 2022). The eight sigma levels - 0.025, 0.095, 0.20, 0.34, 0.51, 0.685,
0.835, and 0.95 - correspond to pressure levels - 25.35 hPa, 96.33 hPa, 202.8 hPa, 344.76 hPa, 517.14 hPa, 694.59 hPa, 846.69





hPa, and 963.3 hPa assuming standard atmospheric pressure of 1013 hPa over the ocean. The model variables are organized into three categories: prognostic, diagnostic, and forcing. The **prognostic variables**, which act as both inputs and outputs, include temperature (T), specific humidity (SH), zonal wind (U), meridional wind (V), and the natural logarithm of surface pressure (logP). With the exception of logP, these variables are defined across all eight vertical levels of the atmosphere. The **diagnostic variable**, used exclusively as output, is total precipitation (TP). The **forcing variables**, used only as inputs, consist of orography, total incoming solar radiation (TISR), land–sea mask, and monthly $CO_2$ concentrations interpolated to a 6-hourly resolution. For simulations following Atmospheric Model Intercomparison Project (AMIP)-style configurations, sea surface temperature (SST) is included as an additional forcing variable. The complete training dataset amounts to approximately 35 GB.

## 3 Methods

The architecture of LUCIE-3D follows the overall design of its two-dimensional predecessor, LUCIE-2D (Guan et al., 2024). The backbone of the model is the Spherical Fourier Neural Operator (SFNO; Bonev et al., 2023), which is particularly well-suited to the spherical geometry of the Earth system and facilitates stable long-term simulations. In this study, we present an updated version of LUCIE-2D that includes an increased number of trainable parameters and additional prognostic variables spanning all $\sigma$-levels of the atmosphere. To accommodate the larger training dataset and expanded set of variables, the model architecture has been extended to 12 layers with a latent dimension of 256.

### 3.1 Integration and Loss Function

LUCIE-3D continues to employ an Euler integration–based constraint to predict the tendencies of the prognostic variables and reconstruct their full fields, similar to LUCIE-2D. During training, the model ingests the full fields of prognostic variables together with the forcing variables at time $t$ as inputs, and outputs the prognostic variables and full-field precipitation at $t + \Delta t$. The loss function remains unchanged from the previous version. Specifically, a Legendre–Gauss quadrature–weighted $L_2$ loss is applied to the prognostic variables, while a standard $L_2$ loss is used for the diagnostic variable. This distinction is necessary because applying the quadrature-weighted $L_2$ loss to precipitation was found to degrade both the stability and the learning of the prognostic variables. Further details of the loss formulation are provided in Section 3 in Guan et al. (2024).

### 3.2 Training Strategy

The training of LUCIE-3D consists of two phases. In the first phase, the model is pre-trained using the quadrature–weighted $L_2$ loss function described above. After the first 20 epochs, a validation-loss-scaled weighting scheme introduced by (Ocampo et al., 2024) is applied and updated every 10 epochs. This scheme computes the weight for each variable's loss as

$$w = \frac{0.005}{\text{validation loss}},$$



which helps prevent the model from overfitting to specific variables, such as specific humidity at lower $\sigma$ - levels or precipitation. To further stabilize training, the relative weights in the loss function for $\log P$ and $TP$ are manually reduced by a factor of 0.5 to avoid excessively large loss values.

Following pre-training, the model enters a fine-tuning phase in which an additional Fourier-based spectral regularizer is incorporated for 30 epochs to correct spectral bias (Guan et al., 2024; Chattopadhyay and Hassanzadeh, 2023). A summary of the key model and training hyperparameters is provided in Table 1.

| Hyperparameter | Value |
| --- | --- |
| SFNO blocks | 12 |
| Encoder & decoder layers | 1 |
| Latent dimension | 256 |
| Maximum learning rate | 5e-4 |
| Minimum learning rate | 1e-8 |
| Batch size | 32 |
| Number of epochs | 160 |
| Optimizer | Adam |
| Activation function | SiLU |
| Regularizer weight | 5e-2 |
| Weight decay | 1e-5 |

**Table 1.** Summary of LUCIE-3D hyperparameters.

## 4 Results

In this section, we evaluate the emulation performance of LUCIE-3D in terms of its ability to reproduce long-term climatological statistics under present-day $CO_2$ forcing, as well as its forced response in AMIP-style climate change simulations with prescribed SST perturbations. We further highlight the improvements achieved by LUCIE-3D over its two-dimensional predecessor in representing large-scale variability, particularly in the precipitation spectra. At the same time, we discuss the challenges that remain for this emulator, including its ability to capture meaningful and simple correlations between variables (e.g., how the sea surface influences low-level atmospheric temperature) as well as the impact of $CO_2$ on stratospheric temperature.

### 4.1 Climatology

Figure 1 presents the climatological biases of LUCIE-3D relative to ERA5 for temperature, zonal wind, and specific humidity as a function of the vertical levels resolved by the model. Consistent with results from LUCIE-2D, the tropospheric climatology is



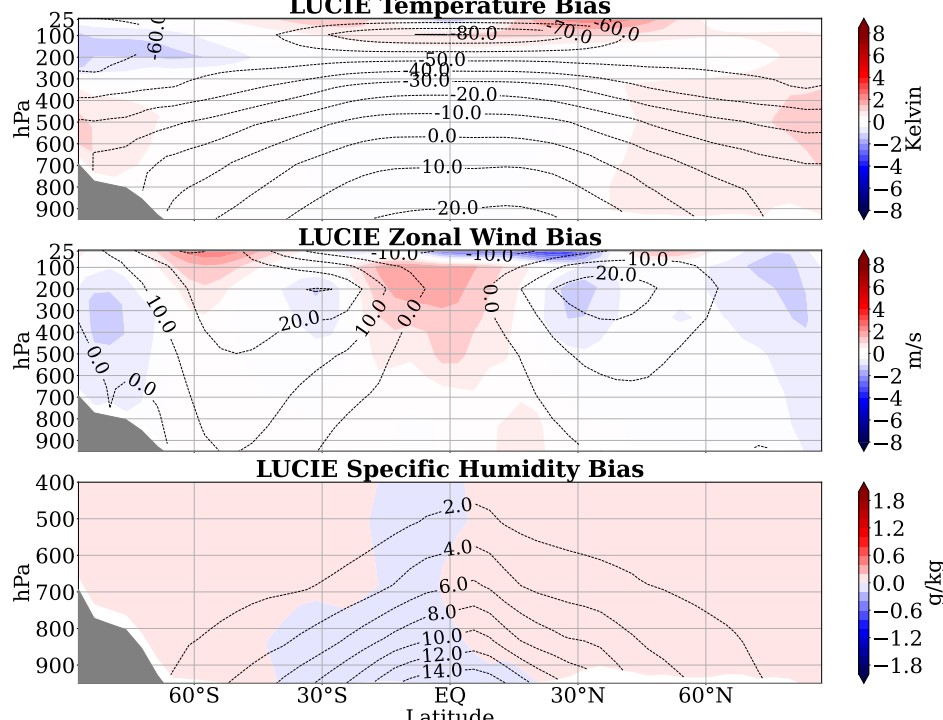

**Figure 1.** Climatology bias of temperature, zonal wind, and specific humidity of LUCIE-3D as compared to ERA5 for the period 1981 - 2020. Results are shown for (top) the temperature (middle) zonal wind, and (bottom) specific humidity with the dash lines representing the climatological values for each variable for ERA5.

captured with good accuracy. This is particularly notable for specific humidity, where LUCIE-3D has little biases even in areas influenced by convection such as the mid-troposphere over the equatorial regions. However, more substantial biases are evident in the stratosphere. Similar difficulties in representing stratospheric processes have also been reported for other fully AI-based climate emulators, including ACE2 (Watt-Meyer et al., 2025) and CAMulator (Chapman et al., 2025). LUCIE-3D's inability to properly simulate the quasi-biennial oscillation (QBO) (both the frequency and overall structure) is the primary driver of the zonal wind and temperature biases in the equatorial stratosphere. Overall, the vertical bias structure is characterized by enhanced errors in the stratosphere and by increased zonal wind biases near the poles.

## 4.2 Climate change

Figure 2 shows the climatological differences in temperature, zonal wind, and specific humidity between 1981–1990 and 2010–2020. The temperature response in LUCIE-3D exhibits stronger overall changes than ERA5, but the spatial patterns are broadly consistent. Both ERA5 and LUCIE-3D capture tropospheric warming at high latitudes (e.g., olar amplification). Although the





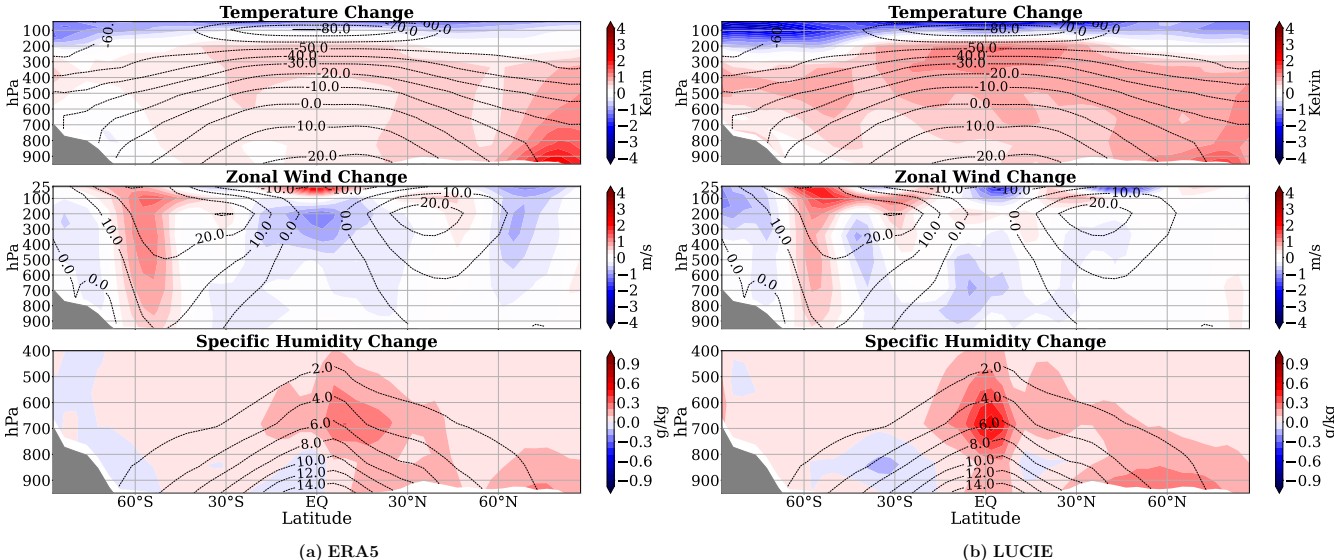

**Figure 2.** Zonal climate change map of (a) ERA5 (b) LUCIE-3D. The dashed contour lines represents the climatology of ERA5 as the reference for the change. The climatology change is calculated as the climatology difference between 1981-1990 and 2010-2020, for both LUCIE-3D and ERA5.

magnitude in LUCIE-3D is somewhat larger, its simulated upper-tropospheric warming in the tropics closely matches ERA5.

In the stratosphere, both LUCIE-3D and ERA5 show a cooling trend. For zonal wind, LUCIE-3D reproduces the large-scale pattern of the strengthening and poleward movement of the jet stream in the Southern Hemisphere, but does not capture the westward changes in the Northern Hemisphere. For specific humidity, LUCIE-3D captures the tropical moistening signal, with a somewhat stronger increase in the lower troposphere over the Northern Hemisphere and a slightly weaker signal in the subtropics. Overall, the magnitude and structure of the differences indicate that LUCIE-3D is able to capture climate-change

responses when $CO_2$ is used as a forcing variable.

To further examine the model's sensitivity to $CO_2$ forcing, Figure 3 illustrates the warming trend of surface temperature in experiments where LUCIE-3D is trained separately with observed $CO_2$ only, and with both $CO_2$ and SST as forcing variables, with each of them evaluated with two scenarios: (1) rising observed $CO_2$ and (2) assuming the values of $CO_2$ every year remain the same as that of the year 1981 (*stationary $CO_2$* from now on). In both surface and stratospheric panels, LUCIE-3D repro-

duces the expected response: under observed $CO_2$, the surface warms and the stratosphere cools, whereas under stationary $CO_2$, temperature trends remain nearly flat. Specifically, the surface temperature trend with real $CO_2$ forcing is +0.20 K decade$^{-1}$, nearly identical to ERA5 (+0.205 K decade$^{-1}$), while the stationary $CO_2$ scenario yields a negligible −0.015 K decade$^{-1}$. With prescribed SSTs, warming remains robust under real $CO_2$ (+0.2146 K decade$^{-1}$) but collapses to +0.0127 K decade$^{-1}$ under stationary $CO_2$. In the stratosphere, LUCIE-3D cools at −0.66 K decade$^{-1}$ (ERA5: −0.47 K decade$^{-1}$), while the stationary





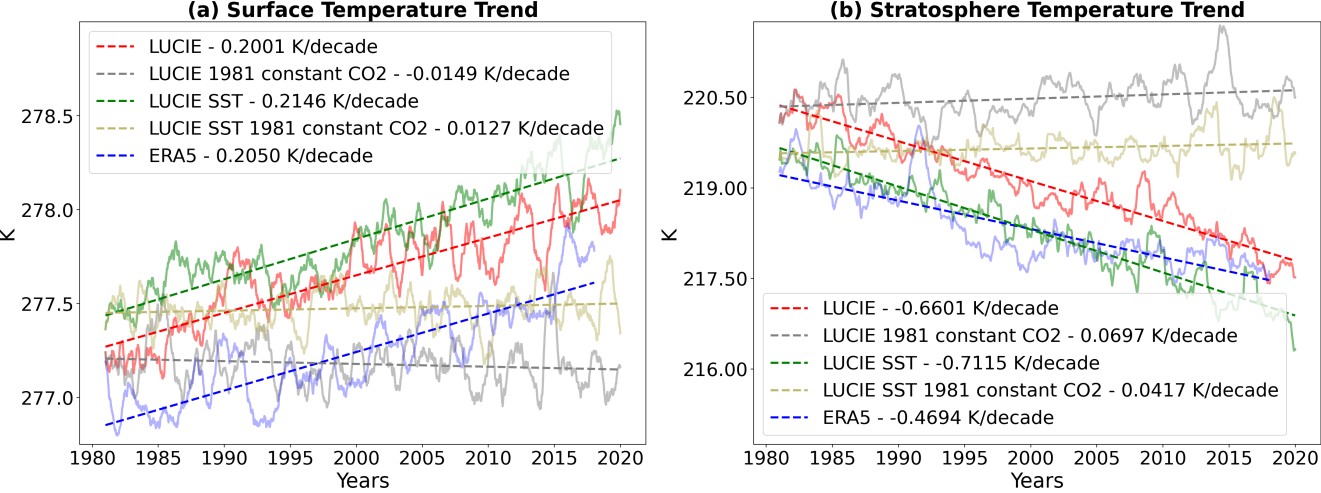

**Figure 3.** Warming trend of (a) surface temperature and (b) stratosphere temperature. LUCIE-3D models are separated into LUCIE-3D and LUCIE-3D trained with SST forcing. Both models are run in inference mode with real $CO_2$ and stationary $CO_2$ (held at the same values as they were in 1981), for 40 years in total.

$CO_2$ case shows a weak positive drift ($+0.0697$ K decade$^{-1}$). Similarly, the SST-only experiment cools at $-0.7115$ K decade$^{-1}$ under real $CO_2$, but remains nearly neutral ($-0.0417$ K decade$^{-1}$) when $CO_2$ is fixed. These results demonstrate that LUCIE-3D has *not memorized* the effect of $CO_2$ but has learned the relationship between the dynamics and the forcing. LUCIE-3D reproduces the surface warming and stratospheric cooling responses to increasing $CO_2$ forcing, while showing negligible long-term changes when $CO_2$ is held constant.

Finally, we test the performance of LUCIE-3D under biased SST forcings by retraining the model with SST fields increased by +2 K and +4 K (applied only over the ocean) in Fig. 4. The model remains numerically stable and physically consistent, producing a warming response over both ocean and Southern Hemisphere land (Fig. 4). However, a spurious cooling trend appears over Northern Hemisphere land. This behavior can be attributed to the use of prescribed SST during training, in which land values are fixed at 270 K, leading to strong land–sea discontinuities. To mitigate this, we apply a smoothing

procedure in which the SST over the ocean and coastal land points are mixed using a Gaussian convolution and normalized by a smoothed ocean mask. This interpolation preserves inland temperatures while reducing artificial discontinuities near coastlines. Retraining the model with the interpolated SST fields improves the response to a +4 K perturbation to SST. It yields consistent warming signals over both oceanic and land regions, without the unrealistic cooling signals over land. However, even with these changes, the model tends to under-predict the degree of warming under the 2+ and 4+K simulations, suggesting there

is still more work needed before emulators are able to extrapolate well outside of their training data. We note this behavior was



also observed in ACE2 and NeuralGCM (Watt-Meyer et al., 2025; Kochkov et al., 2024), where the models failed to respond correctly to 2+ and +4K perturbations of SSTs.



**Figure 4.** Climatology and difference of the 5 year inference of surface temperature, surface specific humidity, zonal wind at $\sigma_{0.95}$ vertical level, and precipitation, with 0K, 2K, and 4K bias in observed SST input.

## 4.3 Variability

The Wheeler–Kiladis diagram is a key diagnostic for assessing the long-term physical consistency of a climate emulator. In particular, the model's ability to reproduce the Madden–Julian Oscillation (MJO), a dominant mode of subseasonal variability in the tropics, demonstrates skill in subseasonal forecasting and in capturing the mechanisms that drive extreme precipitation events in tropical regions. LUCIE-3D closely matches ERA5 in spectral power within the MJO band. Additionally, it success-




**Figure 5.** Climatology and difference of the 5 year inference of surface temperature, surface specific humidity, zonal wind at $\sigma_{0.95}$ vertical level, and precipitation, with 0K, 2K, and 4K bias in interpolated SST input.

fully captures Equatorial Rossby (ER) waves, as the earlier 2D version of LUCIE (Guan et al., 2024). As hypothesized in prior work, incorporating the full vertical structure of the atmosphere in LUCIE-3D grants the model the ability to represent the

spectrum of Kelvin waves.

We further examine the Northern Annular Mode (NAM) and Southern Annular Mode (SAM) in LUCIE-3D, defined respectively as the leading empirical orthogonal function (EOF) of surface pressure over the Northern and Southern Hemispheres. Both NAM and SAM are computed for the boreal winter months: December, January, and February. As shown in Figure 7, LUCIE-3D accurately captures the NAM, with a correlation coefficient of r = 0.95, and reasonably reproduces the SAM with







**Figure 6.** Wheeler-Kiladis diagram of LUCIE-3D and ERA5. Horizontal axis represents the zonal wavenumber ranging from -15 (westward) to +15 (eastward) and the vertical axis represents the frequency. The shading represents the spectral power with long-term climatology. The gray contour lines trace theoretical dispersion relations for equatorially trapped wave modes.

a correlation of r = 0.98. Additionally, the model correctly captures the percentage of variance explained by the leading EOFs of NAM when compared to ERA5 and approximately captured the percentage of variance of SAM.



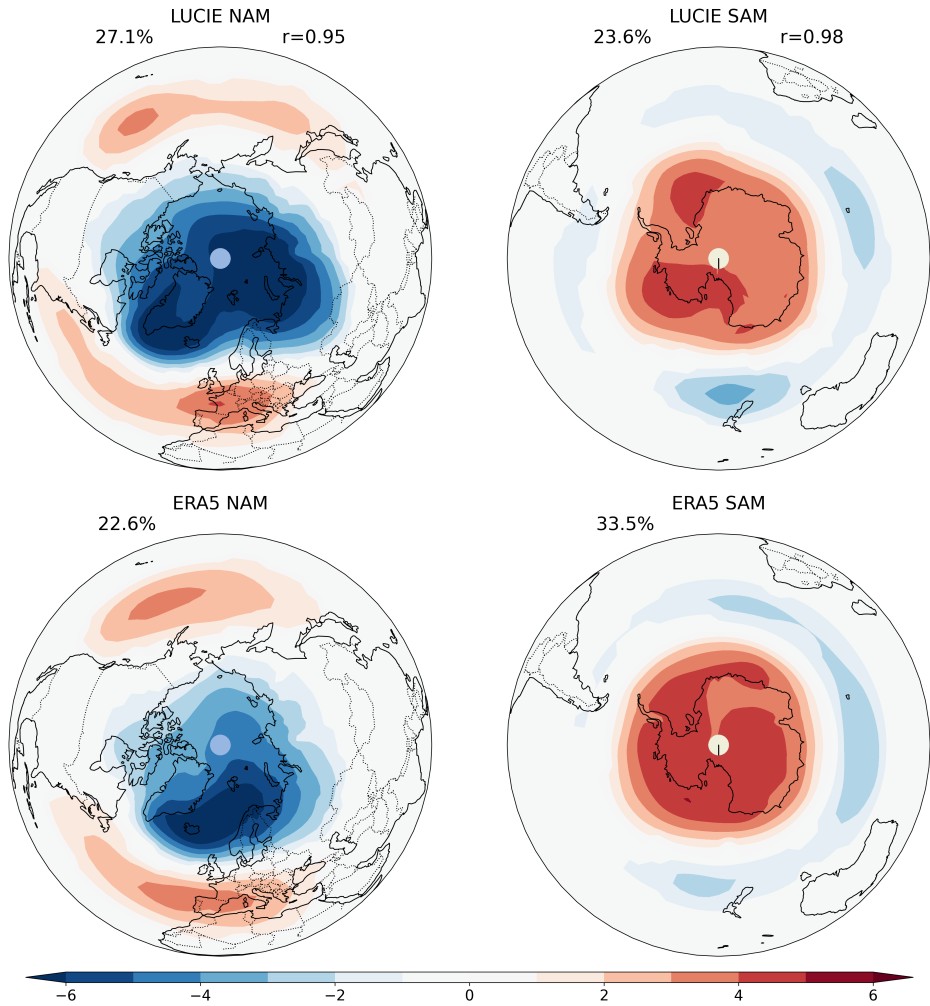

**Figure 7.** Northern Hemisphere Annular Mode (NAM) and Souther Hemisphere Annualr Mode (SAM) of LUCIE-3D and ERA5, calculated with latitude weight $\sqrt{cos(latitude)}$ over the full hemisphere. The percentage represents the first variance fraction and $r$ represents the correlation between LUCIE-3D EOF1 and ERA5 EOF1.

## 4.4 Sudden Stratospheric Warmings

Sudden stratospheric warming (SSW) is a wintertime phenomenon characterized by a rapid increase in polar stratospheric temperatures and a reversal of the zonal-mean winds from westerly to easterly. Due to its irregular occurrence, strong dynamical signal, and ease of diagnosis, SSW serves as a stringent test of the emulator's ability to capture complex, nonlinear atmospheric





behavior. Figure 8 compares ERA5 and LUCIE-3D outputs at 25 hPa. The blue curve denotes the climatological seasonal cycle, the grey shading represents interannual variability, and the red curve highlights a representative SSW winter.

In ERA5, westerly winds strengthen through autumn, then collapse abruptly around mid-winter, accompanied by a sharp rise in stratospheric temperature and a brief transition to easterlies. Here we showcase an example of LUCIE-3D producing 170 one of these events in 2006 with *inference initialized in 1980*, capturing both the rapid wind reversal and the associated sudden warming. While differences remain in the exact timing and amplitude of the event, the emulator captures the essential dynamical signatures of an SSW.

However, this example also highlights a key limitation. Although LUCIE-3D is capable of generating SSW-like events, the timing of these events is not accurately emulated and needs further investigation potentially similar to those performed 175 in Kent et al. (2025).We speculate that as a data-driven model, LUCIE-3D may struggle to fully represent the low-frequency precursors and wave-mean flow interactions that underlie SSW dynamics. These findings suggest that while LUCIE-3D shows promise in emulating major stratospheric events, further work is needed to ensure a robust representation of their variability and underlying physical drivers.

### 4.5 Extremes

Capturing the mean and variability of atmospheric states over long timescales demonstrates LUCIE-3D's capacity for stable inference. However, this alone does not guarantee a statistically robust representation of the climate system. To more comprehensively evaluate the emulator's fidelity, we analyze probability density functions (PDFs) on a logarithmic scale, which is essential for emphasizing rare and extreme events—a prerequisite for credible simulation of climatic extremes.

We assess LUCIE-3D's ability to represent extremes by comparing the logarithms of the PDFs for six key variables against 185 those derived from ERA5 reanalysis: near-surface temperature ($T_{\sigma=0.95}$), specific humidity ($q_{\sigma=0.95}$), upper-tropospheric zonal and meridional winds ($(U, V)_{\sigma\approx0.34}$), surface pressure (SP), and total precipitation (TP). For the dynamical variables—$U$, $V$, and SP—the emulator closely tracks ERA5 across both the bulk and the tails of the distributions, with only minor underestimations at the most extreme quantiles. Similarly, near-surface temperature distributions show strong overall agreement, aside from modest deviations in the warm and cold extremes.

By contrast, the largest discrepancies appear in moisture-related variables. The specific humidity PDF is notably underestimated at its extremities, while the TP distribution exhibits a substantially steeper tail compared to ERA5.

Despite these differences, LUCIE-3D consistently captures the primary features of the target distributions across all variables, indicating a strong capacity to simulate both typical and extreme climate events. However, the underrepresentation of moisture extremes suggests that further refinement may be required to better reproduce rare hydrometeorological phenomena. 195 Difficulties in modeling such "gray-swan" events have been documented in AI-based weather models, and further research is needed to improve their representation in data-driven frameworks (Sun et al., 2024).



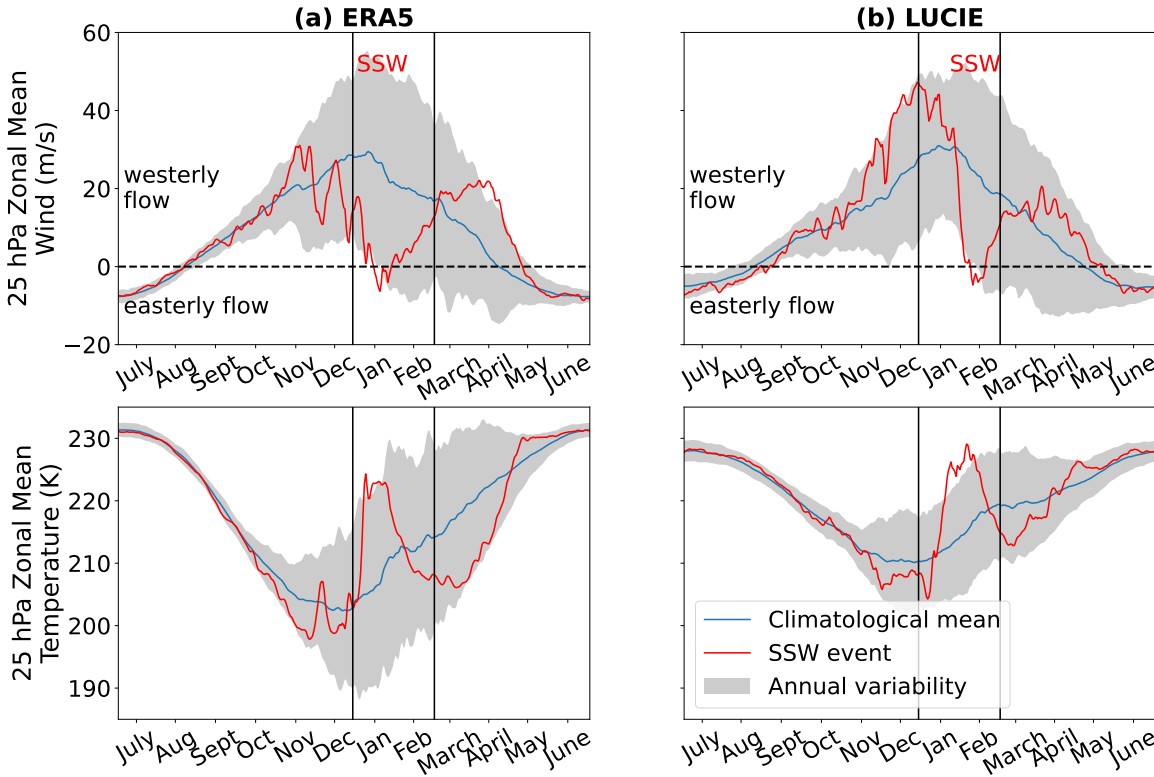

**Figure 8.** The performance of LUCIE-3D in capturing SSW. Results are shown for the (left) ERA5 reanalyses and (right) LUCIE-3D. Results are shown at the top model level ( 25 hPa pressure level) for (top panels) the mean of the zonal wind component in the 55°N–65°N latitude band, and (bottom panels) the mean temperature north of 60°N. Blue curves show the climatological daily mean, while the gray shading characterizes the annual variability by displaying the range between plus and minus two standard deviations. Positive values of the wind indicate westerly flow, while negative values indicate easterly flow. The red curves show the same diagnostics as the blue curves, except for a particular SSW event rather then the 40-year mean. The event from ERA5 took place in 2013.

## 4.6 Spinning up LUCIE-3D from arbitrary climates

In this set of experiments, we perform autoregressive inference with LUCIE-3D by initializing the model with a 30-year climatological mean of the prognostic variables, while using the actual historical forcing variables at each timestep. As shown
in Fig. 10, the model gradually evolves toward the observed climate, capturing both realistic weather patterns and long-term statistical properties. This demonstrates LUCIE-3D's ability to recover from arbitrary initial states and converge toward phys-





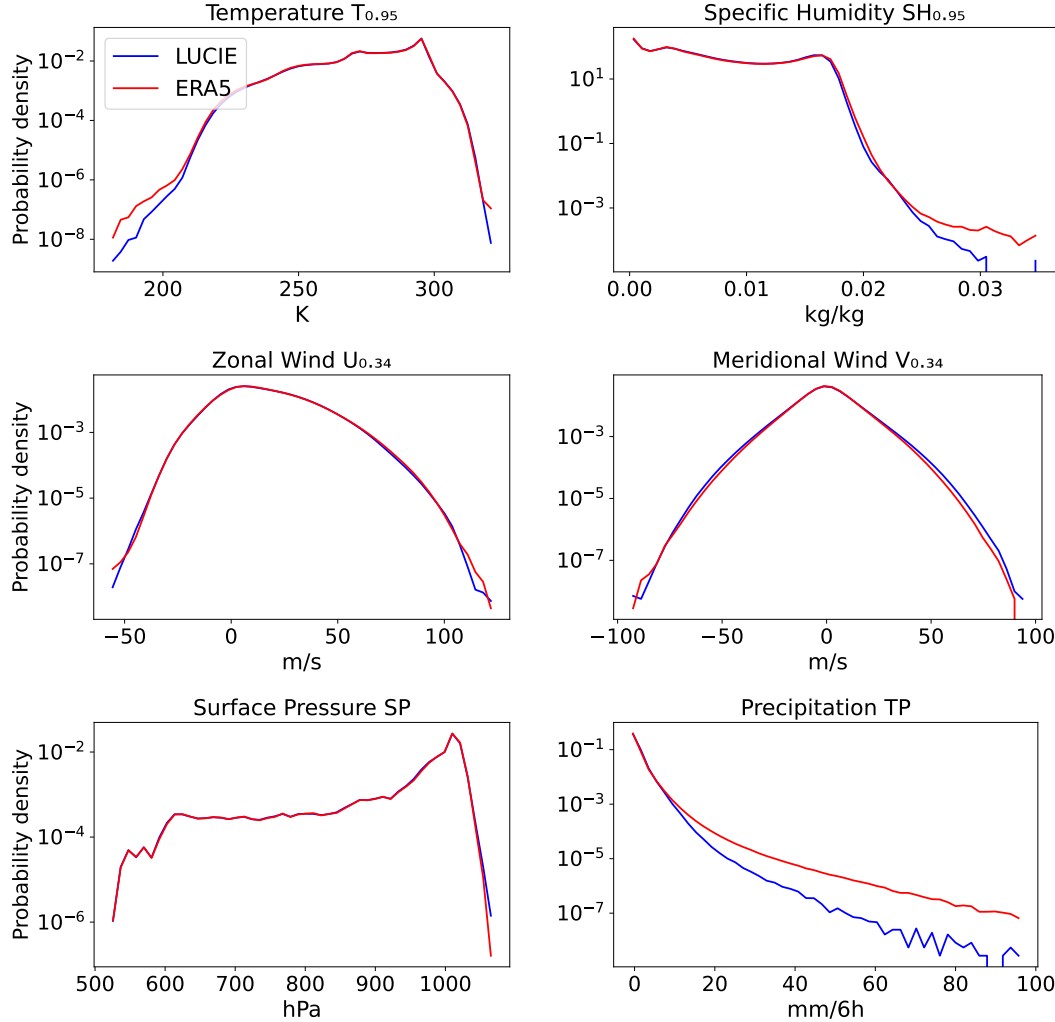

**Figure 9.** Probability density functions (PDFs) at logarithmic scale of the inference of key variables generated by LUCIE-3D, compared to ERA5.

ically consistent atmospheric dynamics. This suggests that LUCIE-3D could potentially be used for idealized dynamical tests and simulations such as those performed in Hakim and Masanam (2024).

To further test the model under more extreme conditions, we conduct an experiment in which LUCIE-3D is initialized with a
"zero atmosphere" (i.e., near-zero values for all prognostic variables) while retaining accurate external forcings. As illustrated in Fig. 11, although the model requires a longer spin-up period, it progressively evolves toward a physically plausible climate state. Figure 12 shows the global-mean surface temperature time series over the first four years of inference for both the clima-





tological and zero initializations. In both cases, initial discrepancies with ERA5 are evident, but LUCIE-3D gradually aligns with the reanalysis data, indicating the emulator's capacity to self-correct from out-of-distribution initial states. This flexibility reduces the model's dependence on carefully specified initial conditions and enhances its utility in broader applications. This test is particularly interesting due to the fact that LUCIE-3D only predicts the tendencies of the prognostic variables, suggesting that the model weights have learned a way to recover to realistic looking atmospheric states (e.g. the imprint of the effects of orography) despite not having that information at the starting initial condition.

To explore the model's sensitivity to external forcing, we perform an additional experiment in which LUCIE-3D is initialized with identical atmospheric states but forced with $CO_2$ concentrations fixed to different historical levels. Specifically, the 1980s simulation uses $CO_2$ data from 1981 repeated annually, while the 2000s simulation uses $CO_2$ data from 2010. As shown in Fig. 13, the resulting surface temperatures are higher in the 2000s simulation, indicating that LUCIE-3D responds appropriately to variations in radiative forcing. This behavior suggests that the model can distinguish climate states characteristic of different eras and may be applicable to paleoclimate or future climate scenarios.

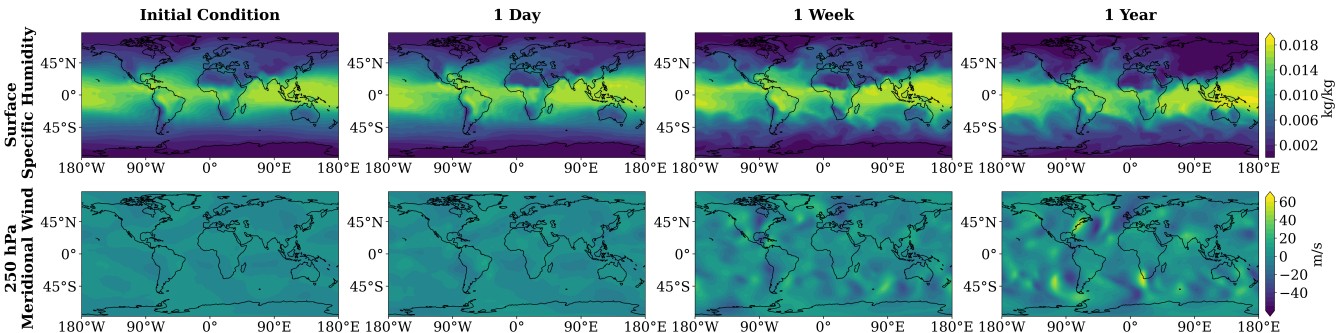

**Figure 10.** Surface specific humidity and model level 2 meridional wind at 1 day, 1 week, and 1 year, with climatology as initial condition.

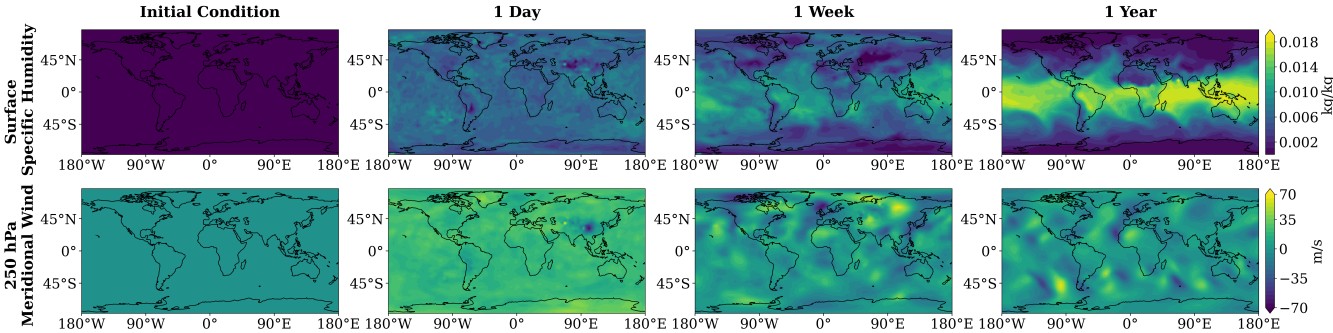

**Figure 11.** Surface specific humidity and model level 2 meridional wind at 1 day, 1 week, and 1 year, with zero as initial condition.




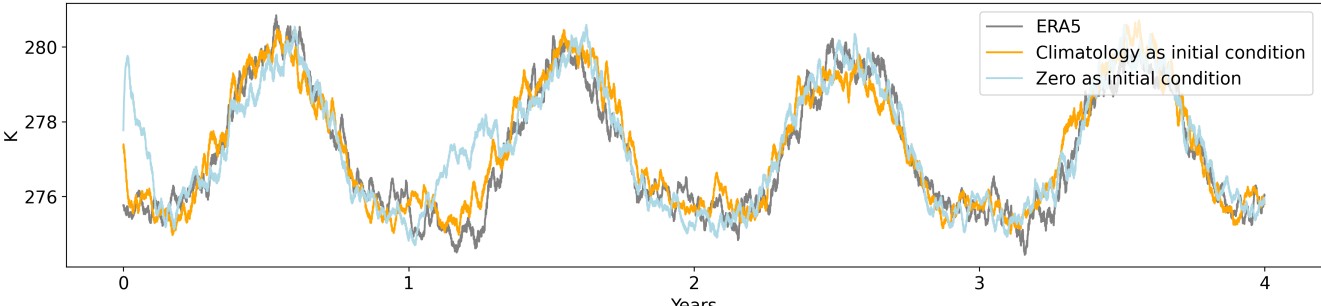

**Figure 12.** Global averaged surface temperature of ERA5, inference with climatology as initial condition, and inference with zero as initial condition.

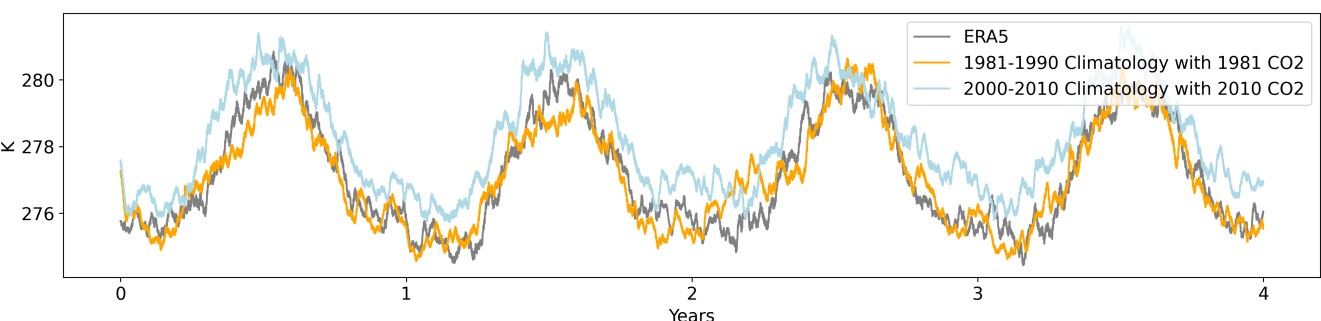

**Figure 13.** Global averaged surface temperature of ERA5, inference with 1981-1990 climatology as initial condition with $CO_2$ repeated in 1981, and with 2000-2010 climatology with $CO_2$ repeated in 2010.

## 5    Discussion

The experiments conducted with LUCIE-3D provide several important insights into the ability of machine learning–based climate emulators to capture long-term forced responses while maintaining computational stability. In particular, the model demonstrates clear skill in reproducing the expected thermodynamic signatures of anthropogenic climate change, namely surface warming and stratospheric cooling under rising $CO_2$ concentrations. These results position LUCIE-3D alongside other recent efforts (Watt-Meyer et al., 2023, 2025; Cresswell-Clay et al., 2024; Kochkov et al., 2024; Chattopadhyay and Hassanzadeh, 2023; Chapman et al., 2025) that have begun to demonstrate robust long-term climate sensitivity in purely data-driven frameworks. Importantly, the separation between the increasing-$CO_2$ and stationary-$CO_2$ experiments confirms that the emulator has learned a physically consistent mapping between radiative forcing and atmospheric response, rather than relying on spurious correlations embedded in the training data. This distinction is particularly valuable for the use of LUCIE-3D in scenario-based studies, as it suggests the model retains sensitivity to external forcings that lie outside the training distribution.



At the same time, it must be noted that since these models are trained on ERA5 data, validation on future climate scenarios is not possible, which is not the case when trained on climate models' outputs, e.g., in Chapman et al. (2025). Furthermore, the accuracy of the responses (in terms of both mean amplitude and structure) of such emulators to new radiative forcings has been debated in recent literature (Van Loon et al., 2025). More focused Green's function experiments (Bloch-Johnson et al., 2024)
should be undertaken in the future with data-driven emulators to understand the challenges and limitations of these models to forcings that are out-of-distribution to the training data.

We further highlight the limitations that remain in the representation of the stratosphere. While the model captures the mean cooling trend and can generate sudden stratospheric warming (SSW)–like events, it does not robustly reproduce the observed frequency or full dynamical complexity of these phenomena. The underlying challenges may be related to the representation of
low-frequency stratospheric variability and wave–mean flow interactions inside data-driven models. The vertical bias structure in LUCIE-3D, which shows enhanced errors in the stratosphere compared to the troposphere, further emphasizes this limitation. Since stratospheric processes such as the QBO and SSWs exert significant influence on surface weather and climate extremes through downward coupling, the incomplete representation of stratospheric variability constrains the applicability of the emulator for studies focused on teleconnections or subseasonal-to-seasonal prediction (Ling et al., 2024; Fernandez et al.,
2024; Peings et al., 2025; Loegel et al., 2025). Several future improvements may involve either targeted training strategies (e.g., stratosphere-specific loss weighting) or the incorporation of hybrid physical constraints that encode known dynamical balances in the stratosphere and reproduce the amplitude and frequency of the QBO.

The spin-up experiments provide a distinct and novel perspective on the stability and adaptability of LUCIE-3D. The emulator was shown to recover from both climatological and extreme "zero atmosphere" initial states, converging toward cli-
matologically consistent conditions under appropriate external forcings. This capability is not trivial since many machine learning emulators exhibit instability or drift when initialized outside their training distribution (Sun et al., 2024). The ability of LUCIE-3D to self-correct demonstrates robustness in its learned representation of atmospheric dynamics and highlights potential utility in paleoclimate contexts, where initial states may be uncertain or poorly constrained (Kadow et al., 2020; Sun et al., 2025; Karamperidou, 2024). Furthermore, sensitivity tests with different fixed-$CO_2$ climatologies (1980s vs. 2000s)
demonstrated that the model consistently evolves toward distinct equilibrated states that reflect the imposed radiative forcing, indicating an appropriate separation of climate states across different eras. This property opens promising avenues for the emulator's application in both paleoclimate reconstruction and projections of future climates under alternative forcing scenarios that may be relevant to geoengineering applications with solar radiation management strategies (Hirasawa et al., 2023; Ahlm et al., 2017; Jones et al., 2009).

The role of ocean–atmosphere coupling in shaping the model response deserves particular attention. In AMIP-style configurations, LUCIE-3D remains numerically stable when perturbed SSTs are prescribed. However, the emergence of spurious cooling over Northern Hemisphere land in the +2 K and +4 K experiments indicates limitations in the emulator's current treatment of boundary discontinuities between land and ocean. This is most striking over the ocean, where the lowest model level temperature does not demonstrate the correct response to the warming SSTs, despite being highly correlated. The smoothing
approach introduced here successfully mitigates this artifact, but it also highlights the limitations of prescribing SST fields

without enabling two-way feedbacks. In reality, coupled dynamics between the ocean and atmosphere play a central role in regulating climate variability, including ENSO, monsoons, and land–sea contrast effects. The present results suggest that while LUCIE-3D can ingest SST forcing and produce physically consistent atmospheric responses, its ultimate fidelity will depend on the incorporation of dynamic ocean–atmosphere coupling. Progress toward such coupling would allow the emulator not

only to represent forced responses but also to generate emergent variability modes associated with air–sea feedbacks, substantially broadening its scope of applicability as shown in some recent work , e.g., in  Wang et al. (2024) and Cresswell-Clay et al. (2025).

Taken together, these results demonstrate both the promise and the present limitations of three-dimensional machine learning climate emulators. LUCIE-3D has advanced beyond its two-dimensional predecessor in reproducing vertical atmospheric

structure, capturing forced responses to $CO_2$, and exhibiting resilience under non-standard initializations. However, the reduced fidelity in the stratosphere, the current lack of fully interactive ocean coupling, and sensitivity to prescribed SST perturbations highlight areas where further development is required. Continued research on hybrid architectures, improved stratospheric representation, and integration of coupled dynamics will be essential to extend the reliability of data-driven emulators across a wider range of climate science applications.

*Code and data availability.*   The training codes, sample emulation, and the trained models is available in this zenodo link: https://doi.org/10.5281/zenodo.17032361 (Guan et al., 2025)

*Author contributions.*   HG conducted the research advised by TA, AC, and RM. All authors analyzed and discussed the results. All authors wrote and revised the manuscript.

*Competing interests.*   The authors declare no competing interests.

*Acknowledgements.*   RM acknowledges support from U.S. Department of Energy, Office of Science, Office of Advanced Scientific Computing Research grant DOE-FOA-2493: "Data-intensive scientific machine learning", and computational support from Penn-State Institute for Computational and Data Sciences. TA was supported by the Global Change Fellowship in the Environmental Science Division at Argonne National Laboratory (grant no. LDRD 2023-0236). AC was supported by the National Science Foundation (grant no. 2425667) and computational support from NSF ACCESS MTH240019 and NCAR CISL UCSC0008.



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
