# Peer review of "LUCIE-3D: A three-dimensional climate emulator for forced responses"

_EGUsphere, 2025_

## Referee Comment (RC2)

Manuscript title: LUCIE-3D: A three-dimensional climate emulator for forced responses

Manuscript number: egusphere-2025-4305

Authors: Guan et al.

My overall recommendation is:

• Major revisions

In brief, my recommendation reflects the need for substantial revisions to address the major and minor comments below. The authors are tackling a genuinely difficult and important problem, and I appreciate the ambition of extending LUCIE into a fully three-dimensional climate emulator. However, in its current form, the manuscript does not yet provide a sufficiently careful or systematic assessment of model performance. Quantitative metrics are used somewhat selectively, some substantial claims are not fully supported, and several aspects of the methods and evaluation are underexplained. It is entirely acceptable for the model to have deficiencies, but these need to be documented and discussed transparently, not only the aspects that work well. With a more thorough and balanced analysis, I believe this work could make a valuable contribution to the ML climate emulator literature.

**Comments to the authors**

**1. Summary of the manuscript**

LUCIE-3D is a lightweight, fully three-dimensional climate emulator designed to capture the vertical structure of the atmosphere while remaining computationally efficient and stable over long integrations. Building on the earlier LUCIE-2D framework, it uses a Spherical Fourier Neural Operator backbone and is trained on 30 years of ERA5 reanalysis data on eight sigma levels. The model takes atmospheric CO2 as an explicit forcing variable and can optionally include prescribed sea surface temperature, allowing it to emulate aspects of coupled ocean–atmosphere dynamics. The authors show that LUCIE-3D reproduces large-scale climatological means, variability and forced climate-change signals, including surface warming and stratospheric cooling under increasing CO2 concentrations. It also captures key dynamical features such as equatorial Kelvin waves, the Madden–Julian Oscillation, and annular modes, and can be spun up from idealized initial conditions. Overall, the study presents LUCIE-3D as an accessible, data-driven framework for efficient exploration of climate responses to external forcing while highlighting its potential for future extensions toward more fully coupled emulation.

The manuscript:

• Introduces LUCIE-3D, a three-dimensional machine learning climate emulator based on a Spherical Fourier Neural Operator backbone.

- Trains the model on ERA5 data (8 sigma levels, 6-hourly, T30 grid) with CO2 (and optionally SST) as forcing.
- Evaluates climatology, forced climate change signals, large-scale variability (e.g. MJO, annular modes), extremes, and spin-up from idealized initial conditions.
- Claims particular novelty in computational efficiency and explicit CO2 forcing, with potential for coupled ocean–atmosphere emulation.

Please find detailed comments below, separated into major and minor points.

**2. Major comments**

**M1. Insufficient detail in the methods section**

The methods section is, in my view, too sparse for a modeling paper in GMD. Key elements of the model design and training procedure are only briefly mentioned or deferred to the LUCIE-2D paper. This makes it difficult for readers to fully understand, reproduce, or adapt LUCIE-3D. I recommend substantially expanding the methods to include:

- A self-contained description of the architecture (input/output variables, SFNO configuration, vertical treatment, and integration scheme).
- Details of the training setup (loss functions, normalization, spectral bias correction, optimizer and learning rate schedule, regularization, and training/validation/test splits).
- A clearer explanation of the Euler integration-based constraint and any other stabilitypromoting design choices.

These additions could partly be placed in an appendix, but the main text should still provide enough detail for the reader to understand the core design and training choices without needing to consult prior work.

**M2. Limited assessment of climatology and model accuracy**

The current evaluation of the model climatology, particularly in Sect. 4.1, feels too limited to support the conclusion that LUCIE-3D achieves "good accuracy". For instance, showing a single vertical structure plot, where positive and negative biases can partially cancel when averaged, is not sufficient to characterize the climatological performance. Saying there is "little bias" doesn't really prove this point. I recommend:

- Expanding the diagnostics to include spatial maps of mean state and biases (at multiple levels), as well as zonal-mean sections, for key variables.
- Providing quantitative metrics (e.g. RMSE, pattern correlation, variance ratios) for climatology over well-defined regions and levels.
- Clarifying the time period over which climatological statistics are computed and ensuring consistency across figures.

A more systematic assessment would better support the claims about climatological accuracy and help readers understand where the emulator performs well and where it has limitations. In addition, the discussion of stratospheric representation and the role of the QBO should be tempered. The model has only a single vertical level within the typical QBO altitude range, and the QBO is primarily a tropical signal. Under these constraints, it is not realistic to attribute stratospheric deficiencies to the absence or misrepresentation of the QBO. Instead, the manuscript should emphasize the limited vertical extent and coarse stratospheric resolution of LUCIE-3D as the primary factors shaping its stratospheric performance and be more cautious in drawing conclusions about QBO-related behavior.

**M3. Limited and selective use of quantitative metrics**

Across the manuscript, model performance is often described using qualitative phrases such as "high accuracy" or "low bias" without accompanying quantitative metrics. In contrast, where the model performs particularly well (e.g., the spatial correlations of the SAM and NAM modes of 0.95 and 0.98, respectively), these strong metrics are highlighted, but even there important caveats are not fully discussed (for instance, the fraction of explained variance differs substantially from ERA5 for each of these modes). It is perfectly acceptable for the model to have deficiencies; however, these should be documented transparently and supported by quantitative diagnostics. I encourage the authors to adopt a more systematic use of metrics throughout (e.g., RMSE, variance ratios, correlation coefficients, explained variance) and to discuss both strengths and weaknesses wherever possible.

**M4. Interpretation of the SSW case and implied predictability**

Section 4.4 shows a single SSW-like event at 25 hPa and states that LUCIE-3D produces "one of these events in 2006 with inference initialized in 1980". As written, this can be read as implying that the model is expected to reproduce the timing of an individual SSW event many months (or even decades) after initialization. Given that SSWs are generally regarded as having limited predictability on subseasonal scales of at most a few weeks (e.g., Cho et al., 2023, and related work on SSW predictability), this raises several questions:

- What exactly is being claimed here in terms of predictability? Is the goal to show qualitative capability to generate SSW-like events under realistic forcing, or to reproduce the timing of specific observed events?
- What aspects of the forcing or model design would make it reasonable for an SSW to occur in both ERA5 and LUCIE-3D in the same winter, given the long lead time from the stated initialization date?
- If the emulator is primarily intended as a climate model (rather than a forecast system), is it appropriate to emphasize the coincidence in calendar year at all, or could this be misread as evidence of overfitting or overly strong imprint of the training data?

I would recommend clarifying the intent of this example and aligning it with current understanding of SSW predictability. For instance, you could frame it more explicitly as a qualitative demonstration that LUCIE-3D can produce SSW-like events with realistic structure, and complement this with a more statistical evaluation (e.g., frequency, seasonality, and basic characteristics of SSW-like events), rather than focusing on a single coincident case. Though, now that it is brought up, I suggest clarifying that this is not evidence of overfitting.

**M5. Physical constraints and position within the ML emulator landscape**

The manuscript does not really discuss physical constraints such as mass, water, and energy conservation, nor how well LUCIE-3D respects these quantities in practice. Other emulators (e.g., CAMulator, ACE2) explicitly include fixers or correction steps to enforce or at least improve physical consistency. It would be very helpful if the authors could:

- Clarify whether LUCIE-3D includes any explicit constraints or postprocessing to address conservation of mass, water, and energy, and if not, provide at least a basic diagnostic assessment of how large the associated drifts or imbalances are over long integrations.
- Discuss how these choices affect the intended use cases for LUCIE-3D (for example, short-term sensitivity experiments versus multi-decadal climate response studies).
- Situate LUCIE-3D more clearly within the broader family of ML climate emulators, in particular relative to models that enforce stronger physical constraints such as CAMulator and ACE2. What niche or role do you envision for LUCIE-3D, given its design choices regarding conservation and physical consistency?

A more explicit discussion of physical constraints and where LUCIE-3D sits in the current emulator landscape would make it much easier for readers to understand when and how this model can be reliably used.

**M6. Conclusions and framing of LUCIE-3D's role**

The concluding paragraphs strike a good balance between highlighting the promise of LUCIE-3D and acknowledging several key limitations (stratospheric fidelity, lack of dynamic ocean coupling, and sensitivity to prescribed SST perturbations). I would, however, encourage a bit more specificity and alignment with the main body of the paper:

- It would be helpful if the conclusion more clearly articulated what LUCIE-3D is currently well suited for (e.g., idealized forced-response experiments, present-day climate sensitivity tests, process studies focusing on large-scale tropospheric structure) versus applications where it is not yet reliable (e.g., teleconnection studies strongly involving the stratosphere, detailed SSW/QBO analyses, fully coupled ocean—atmosphere variability).
- The statement that LUCIE-3D can ingest SST forcing and produce "physically consistent" atmospheric responses feels somewhat strong in light of the issues documented earlier in the paper (e.g., spurious land cooling under SST perturbations, limited vertical coverage in the stratosphere). I suggest softening or qualifying this wording, or explicitly stating in what sense the responses are physically consistent.

• Since the discussion emphasizes the need for hybrid approaches, improved stratospheric representation, and coupled dynamics, the conclusion could briefly connect this to concrete next steps for LUCIE-3D (e.g., adding physical constraints or fixers, targeted stratospheric training, coupling to an ocean emulator), rather than only framing these as generic goals for "data-driven emulators" as a whole.

A slightly sharper and more concrete conclusion along these lines would help readers understand both the genuine progress represented by LUCIE-3D and the realistic limits of what it can currently deliver.

**Technical corrections and typos and places to improve**

- **T1**. Page 1, line 10: Replace "(Pathak et al.)" with "(e.g., Pathak et al.)". There are now many ML-based NWP systems, and adding "e.g." makes clear that this is an illustrative, not exhaustive, citation.
- **T2**. Page 1, line 20: Consider adding a more recent and/or peer-reviewed reference in addition to the arXiv preprint (Chattopadhyay and Hassanzadeh, 2023), given that this preprint is now a few years old.
- **T3**. Page 1, line 24: remove (developed by us) or make a more scholarly statement.
- **T4**. Page 2, line 27: I am not fully convinced by the argument that training cost is prohibitive. While non-trivial, a few days on four GPUs does not seem excessively demanding in the current context and could be better justified or rephrased.
- **T5**. Page 2, line 38: The sentence "Unlike its predecessor, which was trained on a limited number of sigma-levels, LUCIE-3D is trained on data spanning the full vertical extent of the atmosphere" is potentially misleading. Because the vertical information is interpolated and the model does not extend above 25 hPa, this is not the full vertical extent of the atmosphere. Please clarify the actual vertical coverage and rephrase accordingly.
- **T6**. Page 2, line 47: The manuscript states that the model has potential for coupling to dynamical ocean models. However, in its present configuration the model does not include the full set of variables typically required for coupling to dynamic oceans. Please clarify what is meant by "potential for coupling" in this context and specify which additional variables or interfaces would be needed.
- T7. Page 2, line 55: "Vertically interpolated across eight" is ambiguous. Do you mean vertically averaged (as in ACE) or simply interpolated to eight fixed levels? Please clarify the procedure. Also, please give the nominal horizontal resolution in degrees (e.g. T30, approximately 3.75° × 3.75°, corresponding to a Gaussian grid of 96 longitude by 48 latitude points) and briefly comment that this is a relatively coarse resolution.

- **T8**. Page 3, section "Dataset": A small table listing all variables used (including units and number of levels per variable) would improve clarity. Please also explicitly cite the ERA5 dataset. It would be helpful to add a brief motivation for the choice of variables.
- **T9**. Page 3, section "Methods": The statement that SFNO is "well-suited" is rather qualitative. Please provide a more scientific justification (e.g. handling of spherical geometry, spectral properties, ability to capture multi-scale dynamics) and, if possible, support this with references.
- T10. Page 3, section "Methods": The description of the model architecture is very brief and relies heavily on the previous LUCIE-2D paper. It would be helpful to include at least a concise, self-contained description of the core architecture here (potentially moving further details to an appendix) so that the manuscript is readable without constantly referring back to earlier work.
- **T11.** Page 3, section 3.1: The "Euler integration-based constraint" is not clearly explained. Is this simply predicting tendencies and then updating the state with an explicit Euler step? Please provide a precise description. At present, the reader must rely on the LUCIE-2D description to fully understand the setup.
- **T12**. Page 3, line 89: Please clarify how the value 0.005 was chosen (e.g. tuning, prior work, sensitivity tests).
- **T13**. Page 3: Please specify whether and how the data were scaled or normalized before training, and describe the training/validation/test split (time periods, fraction of data, etc.). As written, this part of the methods section feels incomplete.
- **T14**. Page 4, line 90: Please provide more detail on the "corrected spectral bias term": what is the exact form of this correction, and how is it implemented in the loss or architecture?
- T15. Page 5, Figure 1 and related text: The uppermost model level lies near the lower portion of the QBO region, so only a small part of the full QBO structure (approximately 10–70 hPa) is represented, one level (25hpa). Thus, I would not expect the model to capture a realistic QBO vertical structure at all. It would just be a fluctuation at the top of your last line. I think line 105-109 is just not relevant for that conversation, given the fact that you are not near the QBO height. Also, the QBO is tropical feature. This suggests the discussion has not fully accounted for the actual vertical coverage of the model.
- **T16**. line 113: typo "polar amplification".
- **T17**. Section 4.2: I agree with reviewer 1 that the 0.5 degree bias seems high for the surface temperature trend.
- **T18**. Section 4.2, line 115 for all of these findings are you still referring to figure 2? how can you look at polar amplification in a single line plot.

**T19**. Section 4.2, line 117. Specific humidity change is in the wrong location in LUCIE with the trend being focused right on the equator rather than in the northern hemisphere tropics. Adjust this comment

**T20. Interpretation of biased SST experiments and smoothing procedure (lines 135–147)**

The discussion of the +2 K and +4 K SST bias experiments raises several concerns. Despite being a nice result, the statement that the model is "numerically stable and physically consistent" seems too strong in light of the clearly unphysical cooling response over Northern Hemisphere land. I would recommend softening this wording or being more specific about which aspects of the solution are physically consistent.

Second, the attribution of this land cooling to prescribed SST fields with land values fixed at 270 K and associated land—sea discontinuities is plausible, but currently presented without direct supporting evidence. The smoothing procedure, mixing SST over ocean and coastal land via a Gaussian convolution and normalization by a smoothed ocean mask, also feels rather ad hoc and is not described in enough detail to be reproducible (e.g., kernel width, definition of "coastal land" points, and whether this is applied during training only or also at inference). Moreover, blending SST into coastal land points is not physically straightforward, since land "SST" is not a well-defined quantity, so it would be helpful to emphasize that this is a pragmatic numerical fix rather than a physically based boundary treatment. Again, this is a nice result, but feels more of an ad hoc numerical remedy than a fully physical solution.

Finally, the claim that the smoothing "improves the response" and the subsequent conclusion that this behavior indicates a broader difficulty for emulators to "extrapolate well outside of their training data" could be made more cautious. It would strengthen the argument to provide simple quantitative metrics demonstrating the improvement (e.g., pattern correlation or RMSE of the warming pattern) and to frame the extrapolation limitation more narrowly in the context of these particular uniform SST perturbation experiments, rather than as a general statement about all emulators.

- **T21**. Section 4.3, line 149: When introducing the Wheeler–Kiladis diagram and the MJO as a key diagnostic for variability, please add appropriate references (e.g., the original Wheeler–Kiladis paper and foundational MJO references) to support this discussion.
- T22. Line 152: The statement that LUCIE-3D "closely matches ERA5 in spectral power within the MJO band" is qualitative. From Fig. 6 it appears that LUCIE-3D may in fact overestimate power in parts of the MJO band. I suggest either (i) providing a quantitative metric (e.g., power ratio, integrated variance in the MJO box, correlation across the spectrum) to substantiate "closely matches", or (ii) softening the wording to acknowledge any apparent overestimation.
- T23. Lines 153–155: The text states that incorporating the full vertical structure in LUCIE-3D

"grants the model the ability to represent the spectrum of Kelvin waves", in contrast to the earlier 2D version. It would be helpful to clarify what specific deficiency existed in LUCIE-2D (e.g., weaker amplitude, incorrect phase speed, missing parts of the Kelvin band) and to show how LUCIE-3D improves on this with quantitative metrics. Since you highlight MJO and equatorial wave representation in the abstract, this section would benefit from a more systematic analysis, including a brief discussion of remaining deficiencies as well as successes, supported by measurable diagnostics (e.g., power spectra in Kelvin and ER bands, comparison to theoretical dispersion curves).

- T24. Line 175; what is the improvement or strategy in Kent et al. (2025)? briefly summarizing which elements of Kent et al. (2025) you have in mind and how they could be applied to LUCIE-3D (for example, a systematic evaluation of SSW frequency, timing, and composite structure, or an analysis of low-frequency precursors), or removing the reference to Kent et al. (2025) at this point and instead making a more generic statement about the need for a dedicated, quantitative SSW evaluation in future work.
- T25. Section 4.4 would we expect to have the ability to predict a SSW at 6 Months lead time? If not this is evidence of the model being too fit to the data. What is driving the SSW such that it should show up in both ERA5 and LUCIE at the same time (1980) at 6 months lead? This paper does not seem to indicate that they are at all that predictable. https://agupubs.onlinelibrary.wiley.com/doi/10.1029/2023JD039559
- **T26**. Line 202-203. Expand on this sentence or remove.

---

## Author Comment (AC1)

1. **Insufficient detail in the methods section**

   The methods section is, in my view, too sparse for a modeling paper in GMD. Key elements of the model design and training procedure are only briefly mentioned or deferred to the LUCIE-2D paper. This makes it difficult for readers to fully understand, reproduce, or adapt LUCIE-3D. I recommend substantially expanding the methods to include:

   - A self-contained description of the architecture (input/output variables, SFNO configuration, vertical treatment, and integration scheme).
   - Details of the training setup (loss functions, normalization, spectral bias correction, optimizer and learning rate schedule, regularization, and training/validation/test splits).
   - A clearer explanation of the Euler integration-based constraint and any other stability-promoting design choices.

   These additions could partly be placed in an appendix, but the main text should still provide enough detail for the reader to understand the core design and training choices without needing to consult prior work.

**Author's response:** We thank the reviewer for this suggestion. The method section has been updated.

Section 3 and section 3.1 describe the SFNO architecture of LUCIE-3D, with the hyperparameters listed in Table 2. The input and output variables are listed in Table 1 with the pressure levels corresponding to sigma levels noted in the caption.

Section 3.2 describes the integration scheme of LUCIE-3D: the model predicts the increment between the timesteps and reconstructs the full-field prognostic variables to perform autoregressive inference.

Section 3.3 shows the loss functions of LUCIE-3D and the training procedure. Section 3.3.1 shows the first stage of training where LUCIE-3D is trained to perform one-step prediction with adaptive weighting on the training loss values of different variables, based on the validation loss values, after the first 20 epochs of training. Section 3.3.2 shows the FFT spectral regularizer in the final stage to fine tune the model's ability to capture the FFT power spectrum.

Section 3.4 is added to the manuscript to describe the probabilistic LUCIE-3D which shows a major advantage of LUCIE-3D. Because of the lightweight nature of LUCIE-3D, a simple plug-and-go method for uncertainty quantification in the training stage can turn LUCIE-3D into a probabilistic model that fits the chaotic nature of the atmospheric system.

**2. Limited assessment of climatology and model accuracy**

The current evaluation of the model climatology, particularly in Sect. 4.1, feels too limited to support the conclusion that LUCIE-3D achieves "good accuracy". For instance, showing a single vertical structure plot, where positive and negative biases can partially cancel when averaged, is not sufficient to characterize the climatological performance. Saying there is "little bias" doesn't really prove this point. I recommend:

- Expanding the diagnostics to include spatial maps of mean state and biases (at multiple levels), as well as zonal-mean sections, for key variables.
- Providing quantitative metrics (e.g. RMSE, pattern correlation, variance ratios) for climatology over well-defined regions and levels.
- Clarifying the time period over which climatological statistics are computed and ensuring consistency across figures.

A more systematic assessment would better support the claims about climatological accuracy and help readers understand where the emulator performs well and where it has limitations. In addition, the discussion of stratospheric representation and the role of the QBO should be tempered. The model has only a single vertical level within the typical QBO altitude range, and the QBO is primarily a tropical signal. Under these constraints, it is not realistic to attribute stratospheric deficiencies to the absence or misrepresentation of the QBO. Instead, the manuscript should emphasize the limited vertical extent and coarse stratospheric resolution of LUCIE-3D as the primary factors shaping its stratospheric performance and be more cautious in drawing conclusions about QBO-related behavior.

**Author's response:** We thank the reviewer for this suggestion. We added the climatology spatial map into Appendix Section A.
We have added the RMSE table to the manuscript in the climatology section. The time period used in the climatological statistics is noted in the caption of the figures now. In addition to the RMSE table, climatology spatial maps are added into Appendix A for ease to see the performance of the model to capture the climatological pattern.
For the discussion about QBO, we agree that the vertical resolution is too coarse to cover the QBO altitude range. We meant to point out that LUCIE-3D cannot capture the stratospheric dynamics accurately. We have changed the wording in the result section. Figure 1 in this document shows the LUCIE QBO compared to ERA5 QBO.

[Figure]

Figure 1. ERA5 QBO and LUCIE QBO during free run period.

**3. Limited and selective use of quantitative metrics**

Across the manuscript, model performance is often described using qualitative phrases such as "high accuracy" or "low bias" without accompanying quantitative metrics. In contrast, where the model performs particularly well (e.g., the spatial correlations of the SAM and NAM modes of 0.95 and 0.98, respectively), these strong metrics are highlighted, but even there important caveats are not fully discussed (for instance, the fraction of explained variance differs substantially from ERA5 for each of these modes). It is perfectly acceptable for the model to have deficiencies; however, these should be documented transparently and supported by quantitative diagnostics. I encourage the authors to adopt a more systematic use of metrics throughout (e.g., RMSE, variance ratios, correlation coefficients, explained variance) and to discuss both strengths and weaknesses wherever possible.

**Author's response:** We thank the reviewer for this suggestion.
We have added the climatological RMSE table to the Result Section 4.1. With the RMSE values listed for all the variables on different sigma levels, it is clear to see, especially with wind variables, the degrading performance from lower altitude levels to higher altitude levels. Some variables including specific humidity may not be obvious for the small values on the lower sigma levels so we have added climatological spatial maps into the appendix for the reader to clearly see how well the model is capturing the climatology patterns. For all the metrics, we have noted the year range of data used.

4.  **Interpretation of the SSW case and implied predictability**

Section 4.4 shows a single SSW-like event at 25 hPa and states that LUCIE-3D produces "one of these events in 2006 with inference initialized in 1980". As written, this can be read as implying that the model is expected to reproduce the timing of an individual SSW event many months (or even decades) after initialization. Given that SSWs are generally regarded as having limited predictability on subseasonal scales of at most a few weeks (e.g., Cho et al., 2023, and related work on SSW predictability), this raises several questions:

- What exactly is being claimed here in terms of predictability? Is the goal to show qualitative capability to generate SSW-like events under realistic forcing, or to reproduce the timing of specific observed events?
- What aspects of the forcing or model design would make it reasonable for an SSW to occur in both ERA5 and LUCIE-3D in the same winter, given the long lead time from the stated initialization date?
- If the emulator is primarily intended as a climate model (rather than a forecast system), is it appropriate to emphasize the coincidence in calendar year at all, or could this be misread as evidence of overfitting or overly strong imprint of the training data?

I would recommend clarifying the intent of this example and aligning it with current understanding of SSW predictability. For instance, you could frame it more explicitly as a qualitative demonstration that LUCIE-3D can produce SSW-like events with realistic structure, and complement this with a more statistical evaluation (e.g., frequency, seasonality, and basic characteristics of SSW-like events), rather than focusing on a single coincident case. Though, now that it is brought up, I suggest clarifying that this is not evidence of overfitting.

**Author's response:** We thank the author for this comment. In this experiment, LUCIE-3D is able to generate several SSW events throughout the long-term inference period from 1981 to 2020 . The one shown in Figure 8 is one of the SSW events that happened in the free run period. SSW events are considered to have limited predictability on subseasonal scales due the chaotic nature and the purpose of this demonstration is to show that they can be generated (and not predicted). Therefore, this experiment does not aim to show deterministic capability on matching specific events as shown in ERA5 after such a long term of inference. Instead, the purpose of this experiment is to demonstrate the qualitative capability of the model to generate realistic SSW events. We have added a clarification in section 4.4.

5.  **Physical constraints and position within the ML emulator landscape**

The manuscript does not really discuss physical constraints such as mass, water, and energy conservation, nor how well LUCIE-3D respects these quantities in practice. Other emulators (e.g., CAMulator, ACE2) explicitly include fixers or correction steps

to enforce or at least improve physical consistency. It would be very helpful if the authors could:

- Clarify whether LUCIE-3D includes any explicit constraints or postprocessing to address conservation of mass, water, and energy, and if not, provide at least a basic diagnostic assessment of how large the associated drifts or imbalances are over long integrations.
- Discuss how these choices affect the intended use cases for LUCIE-3D (for example, short-term sensitivity experiments versus multi-decadal climate response studies).
- Situate LUCIE-3D more clearly within the broader family of ML climate emulators, in particular relative to models that enforce stronger physical constraints such as CAMulator and ACE2. What niche or role do you envision for LUCIE-3D, given its design choices regarding conservation and physical consistency?

A more explicit discussion of physical constraints and where LUCIE-3D sits in the current emulator landscape would make it much easier for readers to understand when and how this model can be reliably used.

**Author's response:** We thank the reviewer for this comment. LUCIE-3D is specifically designed as a purely data-driven, increment-learning model, distinguishing it from models that utilize explicit physical constraints (e.g., ACE2, CAMulator).
Unlike ACE2, which predicts full fields and employs explicit conservation layers, LUCIE is trained to predict variable increments. Our findings suggest that predicting the increments rather than the full state naturally improves long-term stability and climatological accuracy. LUCIE does not currently include explicit physical constraints for mass or energy. This allows for a more flexible, lightweight architecture. We acknowledge that it may lead to conservation drifts over multi-decadal scales. However, the intention is to keep the model purely data-driven to avoid any potential artificial biases.
LUCIE is designed as an efficient tool for rapid climate sensitivity experiments and exploring atmospheric variability. It is accessible to academic groups with minimal resource requirements. It serves as an efficient and effective platform for both climate studies and architecture design. As evidenced by the new addition to the manuscript of a probabilistic learning mode, the lightweight nature of LUCIE can easily be leveraged for novel deep learning paradigms.

6. **Conclusions and framing of LUCIE-3D's role**

The concluding paragraphs strike a good balance between highlighting the promise of LUCIE-3D and acknowledging several key limitations (stratospheric fidelity, lack of dynamic ocean coupling, and sensitivity to prescribed SST perturbations) I would,

however, encourage a bit more specificity and alignment with the main body of the paper:

- It would be helpful if the conclusion more clearly articulated what LUCIE-3D is currently well suited for (e.g., idealized forced-response experiments, present-day climate sensitivity tests, process studies focusing on large-scale tropospheric structure) versus applications where it is not yet reliable (e.g., teleconnection studies strongly involving the stratosphere, detailed SSW/QBO analyses, fully coupled ocean–atmosphere variability).
- The statement that LUCIE-3D can ingest SST forcing and produce "physically consistent" atmospheric responses feels somewhat strong in light of the issues documented earlier in the paper (e.g., spurious land cooling under SST perturbations, limited vertical coverage in the stratosphere). I suggest softening or qualifying this wording, or explicitly stating in what sense the responses are physically consistent.
- Since the discussion emphasizes the need for hybrid approaches, improved stratospheric representation, and coupled dynamics, the conclusion could briefly connect this to concrete next steps for LUCIE-3D (e.g., adding physical constraints or fixers, targeted stratospheric training, coupling to an ocean emulator), rather than only framing these as generic goals for "data-driven emulators" as a whole.

A slightly sharper and more concrete conclusion along these lines would help readers understand both the genuine progress represented by LUCIE-3D and the realistic limits of what it can currently deliver.

**Author's response:** We thank the reviewer for the suggestion.
We have briefly summarized the reliable use case of LUCIE-3D in the last paragraph of the discussion section. We have also added a statement on the potential future research plans. As mentioned in the last paragraph of the discussion section, the model can benefit from better weighting on the different sigma levels to help the model recognize the lower sigma level variables. The significantly slower dynamics in the, for example, stratosphere makes the model prioritize learning the near surface variables. We have also added the condition to the physical consistency: the model is currently only reliable with realistic SST.

**Technical corrections and typos and places to improve**

**T1.** Page 1, line 10: Replace "(Pathak et al.)" with "(e.g., Pathak et al.)". There are now many ML-based NWP systems, and adding "e.g." makes clear that this is an illustrative, not exhaustive, citation.

**Author's response:** We thank the reviewer for the suggestion. The manuscript has been updated accordingly.

**T2.** Page 1, line 20: Consider adding a more recent and/or peer-reviewed reference in addition to the arXiv preprint (Chattopadhyay and Hassanzadeh, 2023), given that this preprint is now a few years old.

**Author's response:** We thank the reviewer for this comment. We have talked to both of the authors and there is currently no flowing-up paper.

**T3.** Page 1, line 24: remove (developed by us) or make a more scholarly statement.

**Author's response:** We thank the reviewer for the suggestion. The manuscript has been updated accordingly.

**T4.** Page 2, line 27: I am not fully convinced by the argument that training cost is prohibitive. While non-trivial, a few days on four GPUs does not seem excessively demanding in the current context and could be better justified or rephrased.

**Author's response:** We thank the reviewer for this comment. The key philosophy behind this project is to build a lightweight climate emulator that requires as little computing resource as possible. This feature allows LUCIE to be deployed on a single A100 GPU or even on CPUs to generate reliable long-term emulation. We expect LUCIE to lower the barrier to accelerate climate research with machine learning, for any academic groups or even individuals with or without access to clusters. Furthermore, the fast training process enables projects, including our own downstream projects, on architectural designs, training procedure refinement, and experimentation on new techniques (e.g. probabilistic training scheme added to the manuscript).

**T5.** Page 2, line 38: The sentence "Unlike its predecessor, which was trained on a limited number of sigma-levels, LUCIE-3D is trained on data spanning the full vertical extent of the atmosphere" is potentially misleading. Because the vertical information is interpolated and the model does not extend above 25 hPa, this is not the full vertical extent of the atmosphere. Please clarify the actual vertical coverage and rephrase accordingly.

**Author's response:** We thank the reviewer for the suggestion. We agree that the coarse vertical grid does not completely cover the full vertical extent. The manuscript has been updated accordingly.

**T6.** Page 2, line 47: The manuscript states that the model has potential for coupling to dynamical ocean models. However, in its present configuration the model does not include the full set of variables typically required for coupling to dynamic oceans. Please clarify what is meant by "potential for coupling" in this context and specify which additional variables or interfaces would be needed.

**Author's response:** The current setup allows the model to take in sea surface temperature as a forcing variable. LUCIE-3D can be initialized with given SST values and provide an ocean model with surface atmospheric variables for the ocean model to predict the SST values which will be input to future steps of LUCIE-3D inference. However, as shown in

https://www.arxiv.org/abs/2509.12490 , the ocean-coupled version of ACE, the results indicate no major change in the dynamics.

**T7.** Page 2, line 55: "Vertically interpolated across eight" is ambiguous. Do you mean vertically averaged (as in ACE) or simply interpolated to eight fixed levels? Please clarify the procedure. Also, please give the nominal horizontal resolution in degrees (e.g. T30, approximately 3.75°x 3.75°, corresponding to a Gaussian grid of 96 longitude by 48 latitude points) and briefly comment that this is a relatively coarse resolution.

**Author's response:** We thank the reviewer for the suggestion. The vertical gridding was simply interpolated from pressure levels, instead of vertically averaged. The detailed procedure is recorded in https://agupubs.onlinelibrary.wiley.com/doi/full/10.1029/2021MS002712, as cited in the Dataset section.

**T8.** Page 3, section "Dataset": A small table listing all variables used (including units and number of levels per variable) would improve clarity. Please also explicitly cite the ERA5 dataset. It would be helpful to add a brief motivation for the choice of variables.

**Author's response:** We thank the reviewer for the suggestion. The manuscript has been updated accordingly. The variables involved in LUCIE-3D training are based on Arcomano el al. ( 2020) (https://doi.org/10.1029/2020GL087776), Arcomano el al. (2022) (https://doi.org/10.1029/2021MS002712), and Arcomano et al. ( 2023) (https://doi.org/10.1029/2022GL102649) as cited in the manuscript. The original intention was to compare the LUCIE-3D results with the climate model in Arcomano el al. (2022). However, since the ML approach of climate modeling is simpler and more demonstrated now, we cut the discussion on the comparison to the traditional models for brevity.

**T9.** Page 3, section "Methods": The statement that SFNO is "well-suited" is rather qualitative. Please provide a more scientific justification (e.g. handling of spherical geometry, spectral properties, ability to capture multi-scale dynamics) and, if possible, support this with references.

**Author's response:** We thank the reviewer for the suggestion. We have added the reason and advantages of using SFNO for climate emulators in Method Section 3.1. SFNO performs convolutions in spectral space where the coordinates are given by a spherical harmonics basis. This guarantees rotational equivariance and mitigates polar artifacts

**T10.** Page 3, section "Methods": The description of the model architecture is very brief and relies heavily on the previous LUCIE-2D paper. It would be helpful to include at least a concise, self-contained description of the core architecture here (potentially moving further details to an appendix) so that the manuscript is readable without constantly referring back to earlier work.

**Author's response:** We thank the reviewer for the suggestion. As answered for comment 1, we have added details to the method section with self-contained description on the architecture, workflow, input/output variables, and loss functions.

**T11.** Page 3, section 3.1: The "Euler integration-based constraint" is not clearly explained. Is this simply predicting tendencies and then updating the state with an explicit Euler step? Please provide a precise description. At present, the reader must rely on the LUCIE-2D description to fully understand the setup.

**Author's response:** We thank the reviewer for the suggestion. The method section has been updated accordingly. We have added a detailed description on Euler integration in section 3.2.

**T12.** Page 3, line 89: Please clarify how the value 0.005 was chosen (e.g. tuning, prior work, sensitivity tests).

**Author's response:** We thank the reviewer for the suggestion. The method section has been updated accordingly. The value 0.005 has been obtained through hyperparameter tuning to maintain the scaled loss value in the approximately same range as the original loss values without weighting.

**T13.** Page 3: Please specify whether and how the data were scaled or normalized before training, and describe the training/validation/test split (time periods, fraction of data, etc.). As written, this part of the methods section feels incomplete.

**Author's response:** We thank the reviewer for the suggestion. The dataset section has been updated accordingly. The dataset is normalized with the z-score normalization with the mean and standard deviation of each variable.

**T14.** Page 4, line 90: Please provide more detail on the "corrected spectral bias term": what is the exact form of this correction, and how is it implemented in the loss or architecture?

**Author's response:** We thank the reviewer for the suggestion. The method section has been updated accordingly. We have added a detailed description of the loss function for training in section 3.3.2.

**T15.** Page 5, Figure 1 and related text: The uppermost model level lies near the lower portion of the QBO region, so only a small part of the full QBO structure (approximately 10–70 hPa) is represented, one level (25hpa). Thus, I would not expect the model to capture a realistic QBO vertical structure at all. It would just be a fluctuation at the top of your last line. I think line 105-109 is just not relevant for that conversation, given the fact that you are not near the QBO height. Also, the QBO is tropical feature. This suggests the discussion has not fully accounted for the actual vertical coverage of the model.

**Author's response:** We thank the reviewer for this comment. As in the answer for comment 2, we agreed that the vertical resolution is too coarse to cover the QBO altitude range for correct QBO representation. We have modified the discussion in Section 4.1.

**T16.** line 113: typo "polar amplification".

**Author's response:** We thank the reviewer for the correction. The manuscript has been updated accordingly.

**T17.** Section 4.2: I agree with reviewer 1 that the 0.5 degree bias seems high for the surface temperature trend.

**Author's response:** We thank the reviewer for this comment. We believe that the source of this problem lies in the absence of TOA (top-of-the-atmosphere) radiation variable in the diagnosed quantities and thus an absence of an energy budget in the loss function. In future development, we expect to improve the weighting scheme on the loss function and further include additional variables for addressing this bias.

**T18.** Section 4.2, line 115 for all of these findings are you still refering to figure 2? how can you look at polar amplification in a single line plot.

**Author's response:** We thank the reviewer for this comment. The findings are indeed based on Figure 2 in the manuscript. We have added a reference to the figure in the statement.

**T19.** Section 4.2, line 117. Specific humidity change is in the wrong location in LUCIE with the trend being focused right on the equator rather than in the northern hemisphere tropics. Adjust this comment

**Author's response:** We thank the reviewer for this correction. The moistening in LUCIE-3D is indeed centered around the equator. We have changed the statement in the manuscript.

**T20.** Interpretation of biased SST experiments and smoothing procedure (lines 135–147) The discussion of the +2 K and +4 K SST bias experiments raises several concerns. Despite being a nice result, the statement that the model is "numerically stable and physically consistent" seems too strong in light of the clearly unphysical cooling response over Northern Hemisphere land. I would recommend softening this wording or being more specific about which aspects of the solution are physically consistent. Second, the attribution of this land cooling to prescribed SST fields with land values fixed at 270 K and associated land–sea discontinuities is plausible, but currently presented without direct supporting evidence. The smoothing procedure, mixing SST over ocean and coastal land via a Gaussian convolution and normalization by a smoothed ocean mask,also feels rather ad hoc and is not described in enough detail to be reproducible (e.g., kernel width, definition of "coastal land" points, and whether this is applied during training only or also at inference). Moreover, blending SST into coastal land points is not physically straightforward, since land "SST" is not a well-defined quantity, so it would be helpful to emphasize that this is a pragmatic numerical fix rather than a physically based boundary treatment. Again, this is a nice result, but feels more of an ad hoc numerical remedy than a fully physical solution. Finally, the claim that the smoothing "improves the response" and the subsequent conclusion that this behavior indicates a broader difficulty for emulators to "extrapolate well outside of their training data" could be made more cautious. It would strengthen the argument to provide simple quantitative metrics demonstrating the improvement (e.g., pattern correlation or RMSE of the warming pattern) and to frame the extrapolation limitation more narrowly in the context of these particular uniform SST perturbation experiments, rather than as a general statement about all emulators.

**Author's response:** We thank the reviewer for this comment.

The driving logic behind smoothing the SST values into the coastline stems from the forcing variables nature of SST. LUCIE-3D responds to the change in the forcing variables. Increasing the SST values with +2 K and +4 K (while fixing the land values) causes the model to force the prediction to stay unchanged over the land. Since the land values are acting as a constraint on the model prediction and are expected to be ignored by the model, blending the SST value into the land aims to mitigate the sharp spatial derivative that may lead to numerical artifacts. We agree that this approach is more of a pragmatic fix than a physically sound solution. The current setup is a proposed potential solution and our effort on solving this problem and we indeed show accurate signs in the responses for both +2K and +4K SST perturbation. We have added this statement into the manuscript and made the code open-source.

**T21.** Section 4.3, line 149: When introducing the Wheeler–Kiladis diagram and the MJO as a key diagnostic for variability, please add appropriate references (e.g., the original Wheeler–Kiladis paper and foundational MJO references) to support this discussion.

**Author's response:** We thank the reviewer for the correction. We have added the original paper from Dr. Wheeler and Dr. Kiladis to the literature review.

**T22.** Line 152: The statement that LUCIE-3D "closely matches ERA5 in spectral power within the MJO band" is qualitative. From Fig. 6 it appears that LUCIE-3D may in fact overestimate power in parts of the MJO band. I suggest either (i) providing a quantitative metric (e.g., power ratio, integrated variance in the MJO box, correlation across the spectrum) to substantiate "closely matches", or (ii) softening the wording to acknowledge any apparent overestimation.

**Author's response:** We thank the reviewer for this comment. We agree that there is slight overestimation in the MJO box. We have added this statement in the manuscript.

**T23.** Lines 153–155: The text states that incorporating the full vertical structure in LUCIE-3D "grants the model the ability to represent the spectrum of Kelvin waves", in contrast to the earlier 2D version. It would be helpful to clarify what specific deficiency existed in LUCIE-2D (e.g., weaker amplitude, incorrect phase speed, missing parts of the Kelvin band) and to show how LUCIE-3D improves on this with quantitative metrics. Since you highlight MJO and equatorial wave representation in the abstract, this section would benefit from a more systematic analysis, including a brief discussion of remaining deficiencies as well as successes, supported by measurable diagnostics (e.g., power spectra in Kelvin and ER bands, comparison to theoretical dispersion curves).

**Author's response:** We thank the reviewer for this comment. We have added a brief discussion on the deficiency of LUCIE-2D in section 4.3. We have added the power spectra of all the variables into the appendix.

**T24.** Line 175; what is the improvement or strategy in Kent et al.(2025)? briefly summarizing which elements of Kent et al. (2025) you have in mind and how they could be applied to LUCIE-3D (for example, a systematic evaluation of SSW frequency, timing, and composite

structure, or an analysis of low-frequency precursors), or removing the reference to Kent et al. (2025) at this point and instead making a more generic statement about the need for a dedicated, quantitative SSW evaluation in future work.

**Author's response:** We thank the reviewer for this correction. We agree that the original writing was confusing. We have changed the wording and removed the reference to Kent et al. (2025).

**T25.** Section 4.4 would we expect to have the ability to predict a SSW at 6 Months lead time? If not this is evidence of the model being too fit to the data. What is driving the SSW such that it should show up in both ERA5 and LUCIE at the same time (1980) at 6 months lead? This paper does not seem to indicate that they are at all that predictable. https://agupubs.onlinelibrary.wiley.com/doi/10.1029/2023JD039559

**Author's response:** We thank the reviewer for this comment. We clarify that the objective of this experiment is to demonstrate the model's qualitative capability to generate physically realistic SSW events, rather than to achieve deterministic predictive accuracy. Given the chaotic nature of the atmosphere and the limited subseasonal predictability of SSWs, we do not expect a 40-year free-running inference to match the specific timing of events in ERA5. Instead, Figure 8 confirms that LUCIE-3D successfully captures the characteristic dynamics and structural evolution of SSW events as an emergent property of the model. We have reworded this section to better reflect our original point.

**T26.** Line 202-203. Expand on this sentence or remove.

**Author's response:** We thank the reviewer for the suggestion. The manuscript has been updated accordingly.

---

## Author Comment (AC2)

**Major Comments:**

1. In Figure 3, it appears that LUCIE-3d has a temperature bias of 0.5 K in its mean state. LUCIE-3d and LUCIE-3d SST have the right temperature trends, but the absolute global mean temperature is off by ~0.5-0.7 K for the surface. Why does this happen? This problem also exists in GCMs in AMIP, but in ML models, wouldn't the expectation be that this bias does not exist since they are trained on ERA5 itself? NeuralGCM (https://research.google/blog/fast-accurate-climate-modeling-with-neuralgcm/) and ACE2 do not appear to have this mean state bias.

**Author's response:** We thank the reviewer for this comment. We are aware of this problem in LUCIE-3D. We believe that the source of this problem lies in the absence of TOA (top-of-the-atmosphere) radiation variable in the diagnosed quantities and thus an absence of an energy budget in the loss function. We do not see an immediate connection to the similar consequence in GCMs. In future iterations of this model, we are incorporating TOA as a variable as well and we believe that will address this specific problem of an underestimated global mean T2m.

2. Pg. 7 Line 138: Why does the model prescribe land temperatures at 270 K? Land temperatures have a significant diurnal cycle and variability. Prescribing them at 270 K isn't the correct value, and it seems like it would limit the ability to use LUCIE-3D for downstream applications. Either land temperatures should be prognosed (as in ACE2) or they should not be included at all (NeuralGCM), but why set them at 270 K? It also seems that this choice makes the model more brittle. It is worrying that the authors have to apply additional smoothing in order to get a reasonable response to +2K SST, and that without the smoothing, the model's land response is of the wrong sign. (While other emulators have errors with +2K SST, they were not fundamentally of incorrect sign over land for temperature and moisture).

**Author's response:** We thank the reviewer for this question. LUCIE takes atmospheric variables including near surface temperature which is directly correlated to the land temperature. The fixed unrealistic value over the land in SST is simply a mask for the model to recognize and ignore the land when processing sea surface temperature. Since SST is only used as a forcing variable, fixing the land value is not proposed to help the model recover the land temperature but to enlighten the model with SST as a driving force of the atmosphere. We have experimented with mapping the near surface temperature to the land values of the SST and update the values with prediction beside this smoothing technique. We simply observed a better performance with the current setup. +2 K SST is an open research topic and to our best knowledge, ACE2 is indeed showing the wrong sign especially over the land with +2 K and +4 K SST. The current setup is a

proposed potential solution and our effort on solving this problem and we indeed show accurate signs in the responses for both +2K and +4K SST perturbation. In Fig. 1 (of this document) we show a comparison of responses of ACE2 and LUCIE-3D for different SST perturbations.

[Figure]

Figure 1. Responses of LUCIE-3D and ACE2 for different SST perturbations. ACE2 shows incorrect sign in the responses while LUCIE-3D does not.

3.  LUCIE-3D is trained with a spectral regularizer, which they say mitigates spectral bias and point to past literature. In order to validate this claim, I think the authors need to show that the spectra are the same in LUCIE-3D and ERA5. (I think past literature explores this, but it uses different models, not LUCIE-3D itself. ) Based on the PDFs in Fig 8, do the moisture and precip variables still have a spectral bias?

**Author's response:** We thank the reviewer for this comment. Precipitation and specific humidity indeed still have spectral bias, especially in the high wavenumber range. In the LUCIE-2D paper (https://agupubs.onlinelibrary.wiley.com/doi/full/10.1029/2025MS005152), we have performed ablations to show the impact that the spectral regularizer applied to only the extra-tropics has on both the global spectra and long-term temporal bias. The regularizer improves the spectrum but cannot completely erase spectral bias. A complete erasure of spectral bias cannot be guaranteed

in any system with power-law spectra and definitely not with systems having multiple variables with different power-law spectrums. We have shown a mathematical derivation and cause of this in https://arxiv.org/abs/2304.07029. We have now added the surface and near surface variables power spectra plots and stratospheric power spectrum plots into the result section. The rest of the power spectra plots are added into the appendix.

4. Is the precipitation climatology believable, and is the spectra precipitation blurred? Precip is validated in Figure 6 and Figure 9 (bottom right), which shows that the precip tail is heavily underestimated. Given the underestimation in Figure 9, does the climatology of precip have a sufficiently low bias? And are the precipitation results significantly blurred as measured by the spectra?

**Author's response:** We thank the reviewer for this comment. We have added the RMSE table into the result section. Precipitation indeed has a noticeable error in the high wavenumber range. We believe this is due to the diagnostic nature of the precipitation. Also, the precipitation error values are significantly lower than variables like near surface temperature. We expect to implement a better weighting scheme in the future for the loss value of precipitation as well as high altitude variables to balance the loss values.

5. To me, this emulator appears to be very similar to ACE2. They both use the same model architecture (SFNO), they both input atmospheric CO2 forcing in the same way, they both have 8 vertical levels with a 6hour timestep, and they both inherit the same stratospheric biases. The main difference is that ACE2 has more diagnostic variables (e.g. turbulent and radiative fluxes) and trains on 1 degree data. LUCIE is coarser resolution, on a T30 grid (~3.8 degree data). This is likely the major reason why LUCIE trains faster than ACE2. Would it be fair to characterize LUCIE-3D as a coarse-resolution version of ACE2?

**Author's response:** We thank the reviewer for this question. LUCIE and ACE2 are indeed similar to each other but with major differences. LUCIE is trained to predict the tendency of the variables (multiplied by $\Delta t$) and ACE2 is trained to predict the full fields. Our experiments with LUCIE-2D (see https://agupubs.onlinelibrary.wiley.com/doi/full/10.1029/2025MS005152) show improvement on the long term stability and climatology with the current setup. ACE2 incorporates physical constraints including conservation laws in the model. LUCIE is purely data driven with no explicit physical constraints in the model. Furthermore, as newly added into the paper, LUCIE is extended with a probabilistic SFNO formulation thereby allowing for ensemble experiments incorporating epistemic uncertainty.

**Minor Comments:**

1. NeuralGCM displays a property where some initial conditions are stable and others are not. Does LUCIE-3D show signs of instability if initialized with different initial conditions? Or does LUCIE-3D fully solve this problem?

**Author's response:** LUCIE-3D is stable with all the initial conditions given to our best knowledge. As shown in Figure 10 and Figure 11, LUCIE can start with unrealistic initial conditions and correct the inference back to the climatology. A major motivation behind several of the differences between LUCIE and ACE2 comes from our theoretical understanding of instabilities in data-driven autoregressive emulators and thus we are confident that LUCIE is a stable model.

2. In Figures 12 and 13, from the caption, I don't understand what the ERA5 data corresponds to, since the x axis is year 0,1,2,3. Is the ERA5 data initialized at 1981 (is it a climatology?) Furthermore, it would be helpful if the Figure 12 and Figure 13 captions stated explicitly if they correspond to the LUCIE-3D variant with SSTs or without. Is the result robust across these 2 LUCIE 3-D variants?

**Author's response:** We thank the reviewer for the suggestion. The performance of LUCIE-3D and LUCIE-3D with SST are similar in the experiments shown by Figure 12 and Figure 13 and the model shown in the figure is LUCIE-3D **without** SST. The ERA5 global average starts from 1981. Since initialization with climatology and zero fields doesn't present a specific starting year, the x axis is labeled as the year of inference. The caption has now been updated to better reflect this.

3. The authors should clarify the approximate resolution in degrees of the T30 grid. This would be helpful for readers.

**Author's response:** We thank the reviewer for this suggestion. The resolution in degrees is added to the Dataset section.

4. I was confused by the title and think that there should be a clarification in the text. Based on the title, I thought that LUCIE-3D included 3D architectural components, like PanGu's 3D Earth-specific transformer (many of the readers will likely be familiar with PanGu). However, from the Zenodo codebase, I think the authors are adding the vertical fields as additional 2D fields, and the SFNO operates on the 2D fields with spherical harmonic transforms.

**Author's response:** We thank the reviewer for this question. By 3D, we mean that the 3D atmosphere is simulated across multiple vertical levels. While the SFNO does not operate on the 2D fields separately, there are MLP layers in between which enforces mixing across the variables. However, we agree with the reviewer that we do not have a 3D Earth specific transformer. LUCIE, despite that is stable while Pangu is not even in the absence of SST forcing (see the LUCIE 2D paper, see https://agupubs.onlinelibrary.wiley.com/doi/full/10.1029/2025MS005152).

5. Missing citation: I think the paper should cite ArchesWeather and ArchesClimate. These are another class of emulators that is designed to be efficient (and accessible to academic labs).

**Author's response:** We thank the reviewer for this suggestion. We have now added ArchesClimate to our literature review.

6. A surprising result and contribution of this paper is that it shows that models cannot extrapolate out of sample for boundary conditions, but they can extrapolate out of sample for initial conditions (e.g. all zero initial conditions). Do the authors have an explanation for this behavior? It's of course hard to say why because of the overparameterized nature of ML which can be a black box, but if the authors have any reason why this might be the case, I would welcome more discussion on this in the Discussion section.

**Author's response:** We thank the reviewer for this insightful comment. I think whether LUCIE-3D or any other class of observation/reanalysis-trained emulators can generalize to out of sample boundary conditions remain to be seen, although we agree that currently it seems to struggle on many fronts. For example, we know that LUCIE-3D does not invert the sign of the responses over land and overall gets polar amplification. Similarly, there are many other aspects of the response that might be thermodynamically incorrect. This is a fundamental ML problem where out-of-distribution generalization remains a challenge without embedding some notion of the change in the model itself. For the initial conditions, we believe that he forcing variables are helping the model to correct the inference back to the attractor. Based on our experiments with LUCIE-2D, without the forcing variables, especially solar radiation, the model won't be able to maintain stability. In another experiment, when the model is given wrong or unrealistic forcing variables, the inference accommodates itself to match with the frequency of the unrealistic solar radiation, for example.

**Technical correction**

1. Typo: Line 113 on page 5 should be "polar amplification" not "olar amplification"

**Author's response:** We thank the reviewer for the correction.

2. Typo: Figure 7 caption "Annualr" should be "Annular"

**Author's response:** We thank the reviewer for the correction.

3. Sometimes the authors refer to a figure without actually stating the figure number. This is a small thing but it would be helpful to ensure the figures are referred to in the text directly by number. For example, page 8 and 9 should say "Fig. 6" explicitly.

**Author's response:** We thank the reviewer for the suggestion. The manuscript has been updated accordingly.

4. Figure 5 caption itself wasn't sufficiently explanatory. It should clarify that the interpolated SST output refers to the Gaussian convolution to smooth land and sea discontinuities, and it should point to the relevant section in the text.

**Author's response:** We thank the reviewer for the suggestion. The manuscript has been updated accordingly.